# SATA-BENCH: Select All That Apply Benchmark for Multiple Choice Questions

Weijie Xu[1]*, Shixian Cui[1]*, Xi Fang[1]*, Chi Xue[1], Stephanie Eckman[1], Chandan K. Reddy[1]

[1]Amazon

## Abstract

Large language models (LLMs) are increasingly evaluated on single-answer multiple-choice tasks, yet many real-world problems require identifying all correct answers from a set of options. This capability remains underexplored. We introduce SATA-BENCH, the first dedicated benchmark for evaluating LLMs on *Select All That Apply* (SATA) questions across diverse domains, including reading comprehension, law, and biomedicine. Our evaluation of 30 open-source and proprietary models reveals a significant gap: even the strongest model achieves only 41.8% exact match, exposing LLMs' inability to reliably identify all correct answers. We find that this weakness stems from two core challenges: *selection bias* - models favor certain choices regardless of content, and *count bias* — models fail to predict the correct number of answers. To address these issues, we propose *Choice Funnel*, a decoding strategy that combines token debiasing with adaptive thresholding to guide models toward complete and accurate selections. Choice Funnel achieves up to 29% higher exact match than competitive baselines while reducing inference cost by over 64%. Our findings expose fundamental limitations in current LLMs and introduce a new framework for diagnosing and improving multi-answer reasoning. We release SATA-BENCH and Choice Funnel to promote LLM development for robust decision-making in realistic, multi-answer applications.

**Data & Code:** github.com/sata-bench/sata-bench

**Data & Dataset Card:** huggingface.co/datasets/sata-bench/sata-bench

## 1 Introduction

Large Language Models (LLMs) have demonstrated remarkable success across a variety of natural language processing tasks, with multiple-choice question (MCQ) answering emerging as a prominent evaluation setting [40, 55]. However, most LLM benchmarks and training pipelines focus on questions with a single correct answer among a fixed set of options (typically four answer choices). This design choice introduces a structural bias, limiting the models' ability to generalize to more flexible, real-world tasks that require identifying multiple correct answers.

Many real-world scenarios demand such flexibility. For instance, a social media moderator might need to evaluate a post for several types of toxic content—such as threats, hate speech, or offensive language—and accurately tag all applicable categories. Similarly, journalists extracting information from news articles or biomedical researchers annotating scientific papers with multiple relevant subdomains face tasks that extend beyond single-choice frameworks. These are instances of **Select All That Apply (SATA)** questions, where more than four choices are presented and multiple answers are required. Existing LLMs often struggle in such scenarios by inaccurately selecting valid options. Figure 1 illustrates this bias and the shortcomings of even advanced LLMs.

---

*Equal contribution. Correspondence to: weijiexu@amazon.com.

Despite the relevance of SATA-style tasks, there is a lack of standardized benchmarks and evaluation methods tailored to this setting. Existing MCQ benchmarks predominantly assess single-answer selection, leaving a gap in our understanding of how well LLMs perform in multi-answer scenarios. To address these challenges, we introduce **SATA-BENCH**, a new benchmark suite specifically designed to evaluate and enhance LLM performance on SATA tasks. SATA-BENCH is uniquely characterized by multiple correct answers to reflect real-world scenarios, and a diverse set of knowledge-based and reasoning-driven questions. By presenting LLMs with varying numbers

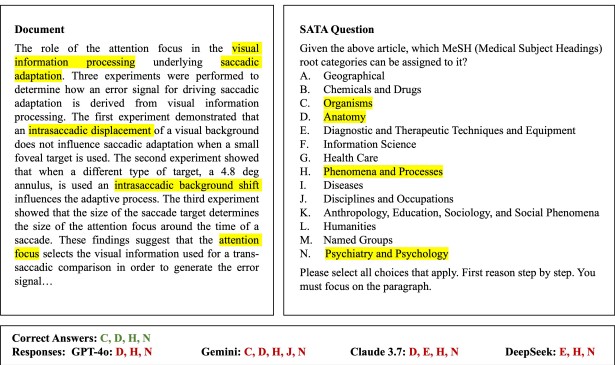

Figure 1: This is a representative example to show that LLMs struggle with SATA (Select All That Apply) questions. The models often provide wrong answers when multiple correct options are present.

of options and multiple valid answers, SATA-BENCH enables the assessment of their ability and biases in more realistic scenarios. SATA-BENCH extends the applicability of LLMs to diverse and complex real-world tasks.

**Our Contributions.** The primary contributions of this paper are:

1. SATA-BENCH *Data Curation*: We curate a high-quality, diverse benchmark dataset explicitly designed to challenge LLMs on multi-answer tasks. SATA-BENCH features 1,604 human-validated questions with varying difficulty levels, multiple correct answers, and carefully constructed distractor options. We provide readability, confusion, and similarity analyses to ensure clarity, diversity, and task complexity across six real-world domains: reading comprehension, news classification, event detection, toxicity identification, biomedical concept tagging, and legal document analysis.
2. *Comprehensive Evaluation of 30 LLMs on SATA Tasks*: We conduct the largest-to-date evaluation of 30 state-of-the-art proprietary and open-source LLMs on SATA questions, revealing that even top-performing models achieve a maximum exact match of only 41.8%. Our analysis highlights two key failure modes—selection bias and count bias—and demonstrates how current LLMs systematically underestimate the number of correct answers. We also break down performance across domains, showing that no single model dominates across all task types, underscoring the diversity and difficulty of SATA-BENCH.
3. *Choice Funnel Decoding Algorithm*: We propose Choice Funnel, a novel iterative decoding strategy that combines token debiasing and adaptive thresholding to mitigate selection and count bias. Choice Funnel outperforms three competitive baselines across 7 open-source models, achieving up to 29% improvements in exact match accuracy while reducing inference costs by over 64%. We demonstrate that Choice Funnel enables smaller models to rival or outperform much larger models on SATA tasks, offering a scalable solution for improving multi-answer reasoning.

## 2  SATA-BENCH Data Curation

Our objective is to develop a dataset that encompasses a diverse range of tasks and domains and poses sufficient challenges to differentiate the capabilities of LLMs. The data curation process consists of three stages: selection of relevant source datasets, transformation of the data into SATA format, and filtering of questions for readability, diversity, and clarity (see Figure 2). We developed SATA-BENCH questions to include tasks in *Reading Comprehension* [28], *Text Classification* (`News` [38], and `Events` [19]), and *Domain Understanding* (`Toxicity` [21], `Biomedicine` [41], and `Laws` [7]). Detailed descriptions of each source dataset are provided in Appendix A.

### 2.1  SATA Transformation

We transformed the original question/text to SATA questions following the steps below: 1. Collect text content, labels, and the total number of options for each question. 2. Maintain the option-to-answer

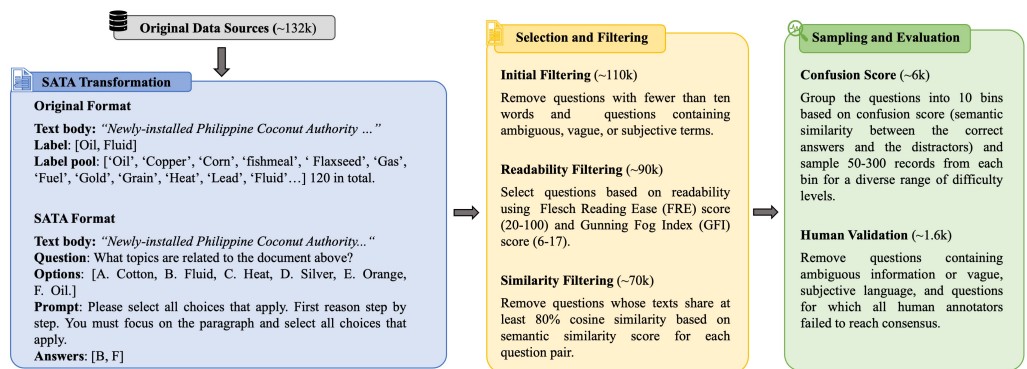

Figure 2: SATA-BENCH Data Curation Process. The source data is converted to SATA format and then filtered for *readability*, *diversity* (via question similarity), *difficulty* (via confusion scoring), and *clarity* (via human validation). Additional dataset-specific transformation steps are described in Appendix B.

Table 1: Statistics of the SATA-BENCH dataset (by data source). We report the following metrics: n: number of instances, LC: label cardinality, m: mean number of correct answers, me: median number of correct answers, min: minimum number of correct answers, max: maximum number of correct answers, r: ratio of the number of choices to the median number of correct answers (LC/me), w: mean word count, FRE: Flesch Reading Ease score, FGL: Flesch-Kincaid Grade Level score, ARI: Automated Readability Index, DCR: Dale-Chall Readability score, GFI: Gunning Fog Index, Confusion: mean confusion score. The final row summarizes these metrics across the entire SATA-BENCH dataset.

| Data Source | n | LC | m | me | min | max | r | w | FRE | FGL | ARI | DCR | GFI | Confusion |
|---|---|---|---|---|---|---|---|---|---|---|---|---|---|---|
| Reading Comprehension | 328 | 3–15 | 2.8 | 2 | 2 | 10 | na | 2018.46 | 59.94 | 9.22 | 12.57 | 9.27 | 9.75 | 0.33 |
| Toxicity | 255 | 8 | 2.56 | 2 | 2 | 6 | 4 | 1015.32 | 37.83 | 12.28 | 13.33 | 10.49 | 12.57 | 0.27 |
| News | 248 | 6 | 2.36 | 2 | 2 | 5 | 3 | 785.93 | 62.51 | 8.92 | 11.15 | 11.1 | 10.94 | 0.26 |
| Biomedicine | 260 | 15 | 5.67 | 5 | 2 | 12 | 3 | 1540.47 | 40.82 | 10.95 | 12.41 | 10.83 | 12.29 | 0.21 |
| Laws | 311 | 15 | 5.3 | 5 | 2 | 10 | 3 | 5761.69 | 45.09 | 12.29 | 14.06 | 8.75 | 12.07 | 0.14 |
| Events | 202 | 6 | 2.63 | 2 | 2 | 5 | 3 | 3644.06 | 50.64 | 10.83 | 13.08 | 9.7 | 11.8 | 0.25 |
| **SATA-BENCH** | **1604** | **3-16** | **3.55** | **3** | **2** | **10** | **3.2** | **2491.01** | **49.56** | **10.75** | **12.80** | **9.96** | **11.51** | **0.24** |

ratio between 2 and 3 for consistency and improved difficulty [50], and determine the number of options ($k$) based on the number of correct answers for each question. 3. Generate distractor options for each question, consisting of the $c$ correct options and $k − c$ options randomly selected from the pool. 4. Shuffle the list of options to eliminate any label imbalance.

## 2.2 Question Filtering

From the original SATA questions (characteristics shown in Table 5 in Appendix), we filter them using the following steps (see Figure 2):

**Initial Filtering.** To clean the original source data, questions with fewer than ten words were eliminated [43, 26]. Additionally, to ensure each question is understandable and solvable, we excluded those containing ambiguous, vague, or subjective terms (details are provided in Appendix B.1) [33].

**Readability.** To ensure SATA-BENCH questions are understandable and challenging, we assessed the readability of each question using the Flesch Reading Ease (FRE) score [20] and the Gunning Fog Index (GFI) [22]. We retained questions with an FRE score between 20-100 (inclusive), filtering out extremely easy or difficult questions [29], and a GFI score between 6-17, corresponding to $6^{th}$ grade to graduate level difficulty [22]. This filtering removed unclear or trivial questions while maintaining a range of difficulties. [2]

---

[2]We also included four additional measures of readability (Flesch-Kincaid Grade Level (FGL) [29], Automated Readability Index (ARI) [29], and Dale Chall Readability (DCR) [12]) in the SATA-BENCH dataset.

**Question Similarity.** Some of the original questions were too similar to each other, adding redundancy to the dataset. Following [57], we calculated the similarity score for each question pair using the cosine similarity score of the term frequency-inverse document frequency (TF-IDF) matrix [46] due to its efficiency. We removed questions sharing at least 80% cosine similarity to reduce duplication.

**Confusion Score.** The difficulty of SATA questions is largely related to how confusing the provided options are. We assessed the semantic similarity between the correct answers and the distractors as a measure of question-level confusion score [6]. We used ST5-XXL [36] for semantic similarity calculation, as it performed best in [34]. Then, to balance the confusion level, we grouped the questions into 10 bins according to the confusion score and sampled between 50-300 records from each bin to ensure SATA-BENCH contains a diverse range of difficulty levels. Figures 5 and 6 show the distribution of the confusion scores before and after filtering as well as by source dataset. We release a dataset comprising 7,983 pre-validation questions as a by-product of our work.

**Human Validation.** Human evaluation was conducted in two stages. First, we used it to identify and remove questions containing ambiguous information as detailed in Appendix B.2. In the final stage, three human annotators reviewed all remaining questions to correct labeling errors; questions lacking consensus among all three annotators were excluded, as detailed in Appendix B.3. Statistics of the final SATA-BENCH is shown in Table 1.

### 2.3 SATA-BENCH Characteristics

After these curation steps, SATA-BENCH has the following characteristics: **(i) Multiple choices with multiple correct answers.** All questions have multiple choices and more than one correct answers. **(ii) Diversity.** The questions come from various disciplines, including knowledge-based tasks such as domain understanding and reasoning-driven questions such as reading comprehension (see Figure 3). **(iii) Human validation.** All questions are manually validated to ensure they are clear and correct by readability scores and human validation. SATA-BENCH removes ambiguous or overly simplistic items while maintaining a range of reading levels (from $6^{th}$ grade to graduate level) (see Figure 3). **(iv) Challenging.** We constructed the SATA-BENCH benchmark so that 76% have a FRE score within the standard range (60-70) and the average GFI score is approximately $13^{th}$ grade (equivalent to the first year of college/university). The mean semantic similarity between correct answers and distractors (incorrect options) is 0.24, exhibiting a right-skewed distribution (skewness = 1.8). Most questions cluster around a similarity score of 0.22, with a few more difficult questions extending the tail towards the higher end (see Figure 3).

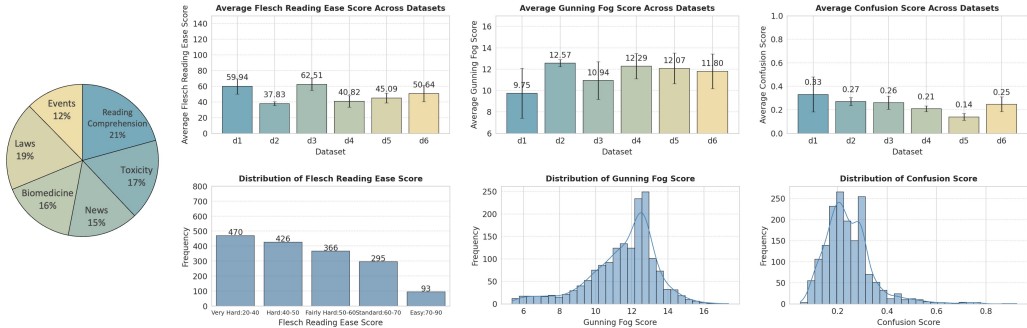

Figure 3: SATA-BENCH Dataset Overview. SATA-BENCH covers a diverse set of topics and achieves a balance between readability and difficulty (measured by confusion score). d1: Reading Comprehension, d2: Toxicity, d3: News, d4: Biomedicine, d5: Laws, and d6: Events.

## 3 Experiments

This section presents the experiments conducted to assess the capabilities of LLMs on SATA questions. The benchmark includes 16 proprietary models and 14 open-source models. (See Table 7 for full model cards.)

**Experimental Setup.** Because our benchmark contains diverse questions, we use a zero-shot evaluation. The system prompt specifies that each question has at least two correct answers, and we

instructed the LLM to output the labeled result in JSONL format [25, 56]. Furthermore, we utilize a CoT prompting strategy as described in [37]. We then extract the answer from the JSONL file using both exact match and fuzzy match. For cases where JSONL extraction fails (fewer than 3% of cases), we use Claude 3 Haiku and Human Labelers to extract the correct options from the answers provided. For smaller models, the percentage of cases where JSONL extraction fails exceeds 5%, making the above methods less reliable. Following [24], we remove CoT and use the probability of the first output token to retrieve options. We hold out a dataset of 100 randomly sampled instances from the benchmark dataset to generate a threshold for each model with optimal Jaccard Index [5]. We select all options with a probability greater than that threshold value. Note that this method applies only to models with accessible token probabilities. We have also included the performance of non-expert humans on the benchmark (see Appendix E).

**Metrics.** We evaluate models using metrics across three categories: performance, selection bias, and count bias (details in Appendix F). Performance is measured using Exact Match (EM), Jaccard Index (JI), Mean Average Precision (Precision), and Mean Average Recall (Recall). Selection bias includes RStd [55] and RSD [11, 42]. We also introduce Selection Probability Divergence (SPD) to quantify unselection bias, a form of selection bias where models consistently avoid certain options. Count bias is assessed using the mean count difference (CtDif), mean absolute count difference (CtDifAbs), and count accuracy (CtAcc).

Table 2: Performance comparison of 30 different LLMs across various metrics on SATA-BENCH. We highlight the best (**bold**) and second-best (underline) values. Columns labeled [(↑)] indicate higher-is-better; columns labeled [(↓)] indicate lower-is-better. Models with explicit reasoning capabilities are highlighted in *italic*. All numeric values are rounded to two decimal places. We retrieve exact labels for models evaluated using Inference-Based Retrieval + CoT prompting. For models evaluated under Probability-Based Retrieval, we select labels based on token probability thresholds.

| Model Name | Performance | | | | Selection Bias | | | Count Bias | | |
|---|---|---|---|---|---|---|---|---|---|---|
| | EM↑ | Precision↑ | Recall↑ | JI↑ | SPD↓ | RStd↓ | RSD↓ | CtDif | CtDifAbs↓ | CtAcc↑ |
| *Inference Based Retrieval + CoT* | | | | | | | | | | |
| *O3* | **41.77** | 87.50 | 81.22 | 73.91 | 0.38 | 6.79 | 0.06 | -0.39 | 0.94 | 46.12 |
| GPT4.1 | 40.49 | 85.52 | 85.66 | **75.23** | 0.13 | 5.98 | 0.06 | -0.04 | 0.85 | 45.52 |
| *Grok 3 Think* | 39.71 | 83.93 | **86.31** | 74.40 | 0.30 | 6.26 | 0.07 | 0.06 | 0.93 | 44.24 |
| GPT4 | 39.47 | 85.90 | 83.17 | 74.11 | 0.21 | 6.63 | 0.06 | -0.20 | **0.82** | 46.61 |
| *Claude 3.7 Think* | 37.92 | 85.03 | 78.77 | 70.96 | 0.46 | 18.77 | 0.34 | -0.32 | 0.87 | 44.48 |
| Claude 3.7 | 37.82 | 85.35 | 77.15 | 70.98 | 0.49 | 6.59 | 0.25 | -0.43 | 0.93 | 43.58 |
| Claude 3 Sonnet | 36.49 | 84.58 | 78.81 | 70.72 | 0.36 | 7.37 | 0.07 | -0.35 | 0.83 | **48.00** |
| *Geimini 2.5 Think* | 36.46 | 84.58 | 83.25 | 72.58 | **0.12** | **4.76** | 0.06 | -0.01 | 0.88 | 43.76 |
| Claude 3.5 Haiku | 35.89 | 80.26 | 85.08 | 71.12 | 0.33 | 7.31 | 0.35 | 0.18 | 1.01 | 42.61 |
| Claude 3 Haiku | 35.64 | 83.59 | 80.16 | 70.63 | 0.42 | 6.24 | 0.07 | -0.22 | 0.85 | 47.15 |
| Claude 3 Opus | 35.59 | 86.97 | 77.19 | 70.15 | 0.62 | 8.26 | 0.07 | -0.52 | 0.93 | 44.36 |
| Gemini 2 Flash | 34.60 | 85.01 | 79.98 | 70.71 | 0.17 | 6.14 | 0.06 | -0.23 | 0.91 | 39.94 |
| GPT 4.1 mini | 33.46 | 86.05 | 78.23 | 69.90 | 0.30 | 6.69 | 0.06 | -0.39 | 0.97 | 38.61 |
| Nova Pro | 32.95 | 87.37 | 75.94 | 68.92 | 0.52 | 7.92 | 0.07 | -0.55 | 1.01 | 39.27 |
| Claude 3.5 Sonnet | 32.22 | 87.57 | 75.25 | 67.15 | 0.43 | 8.41 | 0.09 | -0.46 | 1.06 | 38.55 |
| Llama 3.1 405B | 30.17 | 86.24 | 75.31 | 67.18 | 0.33 | 6.90 | 0.45 | -0.39 | 1.02 | 36.30 |
| Nova Lite | 29.11 | 82.51 | 72.42 | 63.75 | 0.52 | 9.12 | 0.45 | -0.51 | 1.17 | 37.39 |
| *Deepseek R1* | 28.17 | 84.62 | 72.36 | 64.49 | 0.94 | 17.44 | **0.03** | -0.57 | 1.13 | 33.52 |
| Mistral Large V2 | 22.83 | 88.20 | 62.59 | 57.16 | 1.33 | 10.89 | 0.12 | -1.10 | 1.47 | 27.27 |
| Qwen Plus | 21.12 | 88.54 | 59.53 | 55.74 | 2.24 | 10.72 | 0.11 | -1.18 | 1.43 | 24.85 |
| Nova Micro | 18.37 | 86.06 | 60.99 | 55.77 | 1.84 | 11.10 | 0.27 | -1.09 | 1.41 | 24.30 |
| Llama 3.2 90B | 18.30 | **89.56** | 60.80 | 55.78 | 1.84 | 11.10 | 0.27 | -1.09 | 1.41 | 24.30 |
| Llama 3.1 70B | 17.94 | **89.56** | 60.64 | 55.59 | 1.81 | 10.06 | 0.10 | -1.12 | 1.48 | 22.12 |
| Non-expert Human | 17.93 | 60.62 | 54.44 | 45.02 | 1.46 | 15.32 | 1.46 | -0.6 | 1.44 | 34.12 |
| *Probability Based Retrieval* | | | | | | | | | | |
| Mistral 8B | **14.73** | 81.46 | **53.23** | **46.63** | **11.42** | 19.47 | 1.27 | -1.35 | 1.95 | 21.01 |
| Llama3 8B | 13.82 | 80.30 | 47.37 | 43.64 | 12.09 | 17.85 | 1.09 | -1.59 | 1.88 | **22.00** |
| Bloomz 7B | 11.27 | 66.09 | 50.80 | 41.15 | 20.62 | 29.00 | 1.51 | **-0.87** | **1.71** | 20.09 |
| *DeepSeek R1 Distill 8B* | 8.85 | 72.20 | 45.81 | 40.02 | 13.38 | 21.62 | 1.14 | -1.29 | 1.75 | 20.42 |
| Qwen2.5 14B | 6.30 | 87.84 | 38.76 | 37.58 | 21.01 | 18.02 | **1.06** | -2.24 | 2.26 | 11.93 |
| Phi3 7B | 2.97 | **87.25** | 35.67 | 34.57 | 23.22 | 18.57 | 1.22 | -2.33 | 2.35 | 7.22 |
| *Phi4-mini-reasoning* | 2.12 | 77.98 | 30.82 | 29.69 | 21.62 | **13.90** | 1.59 | -2.37 | 2.39 | 7.35 |

## 3.1 Key Observations

**SATA-BENCH is challenging and different.** All models have a precision greater than 80%, but none achieves an EM score above 42%. This indicates that while models often select some correct answers, they fail to consistently identify all of them.

In general, proprietary models have higher EM and Precision than their open-source counterparts. Unlike in other benchmarks, there is no single model that dominates across all performance metrics. Some large reasoning models (LRM), such as O3 and Grok 3 Think, have higher EM and Recall than non-reasoning models. Interestingly, larger and more recent models do not necessarily perform better. For example, Claude 3 Sonnet performs better than Claude 3.5 Sonnet V1 and Claude 3 Opus in exact match rate. However, within the Claude family, larger models always have higher precision. For example, Claude 3 Opus has the highest precision among the Claude 3 model family. According to [4, 13], these results contrast with existing single-choice MCQ LLM performance, such as MMLU [24] and ARC [10], where larger or more recent models tend to show clear improvements.

**Models choose too few answers.** Nearly all LLMs tend to select fewer answers than required. As an extreme example, Llama 3.1 70B on average, selects one fewer option per question than the correct number. Accordingly, Llama 3.1 70B achieves the highest precision but the lowest exact match (EM). The tendency to under-select increases as the number of correct answers grows (see Figure 11). This behavior negatively impacts the EM rate for questions with many correct choices (see Figure 12). The highest CtAcc is only $48\%$, even the best model predicts the correct number of answers in fewer than half of the questions. We hypothesize that this behavior results from models being primarily trained and evaluated on benchmarks where each question has only one correct answer, making them poorly suited for SATA tasks. Through a t-test, we observe that the mean of the CtDif column is significantly lower than 0, with a p-value of $1.70 \times 10^{-6}$, supporting the observation that models consistently under-select answers.

**Unselection bias exists.** Some models exhibit a tendency to avoid selecting specific labels, even when they are correct.[3] When comparing Selection Probability Divergence (SPD) from our benchmark with 1,000 randomly simulated SPDs, Welch's t-test reveals that LLMs' SPD is significantly higher than random, with a p-value of $0.0467$. Even the best model in terms of selection bias (Gemini) underperforms on label M, with its recall rate being 6.3% lower than its average recall (Figure 10).

**There is no clear winner across datasets.** When breaking down the benchmark by its six datasets, different models excel in different domains. Top-performing models vary by dataset, showcasing the diversity of SATA-BENCH. Considering exact match rate (see Figure 13): O3 excels in News and Events classification. DeepSeek R1 leads in Biomedicine, GPT-4.1 performs best in reading comprehension, and Claude 3.7 dominates in toxicity and laws. *This highlights the importance of domain-specific evaluation and the breadth of challenges covered by the benchmark.*

## 3.2 Ablation Studies

We conducted ablation studies to test different strategies for improving model performance. We report the average results across three models selected for diverse profiles in terms of cost, open-source availability, and overall performance. The complete prompts are provided in Appendix H.3.

**Improving performance on SATA-BENCH is challenging.** We tried several approaches to improve performance, but none yielded consistent or significant improvements.

Table 3: Average performance of three models: Nova Pro, Llama 3.1 405B, and Claude Haiku 3.5. The first column shows row numbers for reference.

| | Experiment | EM | Precision | RStd | CtDif |
|---|---|---|---|---|---|
| 1 | 1/2/3/4 | 35.50 | 82.99 | 10.22 | -0.37 |
| 2 | a/b/c/d | 30.69 | 83.10 | 11.56 | -0.26 |
| 3 | default | 33.00 | 84.62 | 7.37 | -0.25 |
| 4 | few shots | 28.35 | 76.61 | 17.33 | -0.42 |
| 5 | option by option | 30.50 | 86.28 | 4.81 | -0.64 |
| 6 | option few shots | 30.87 | 85.80 | 7.93 | -0.48 |
| 7 | with avg count | 27.33 | 76.17 | 14.90 | -0.40 |
| 8 | with count number | 53.95 | 83.30 | 3.45 | -0.08 |
| 9 | single choice | 45.53 | NA | NA | NA |

- **Changing the symbol** used for each answer choice did not improve the selection bias. We replaced the default option IDs from A/B/C/D to a/b/c/d and 1/2/3/4. While the 1/2/3/4 format achieved slightly better exact match accuracy, it also increased selection bias and reduced precision. Overall, we did not observe performance improvement by changing symbols (see rows 1-3 in Table 3).
- We provided **few-shot examples** in the prompt before the test models. However, this strategy did not lead to meaningful improvements in performance (see row 4 in Table 3).

---

[3]However, we cannot exclude position effect [55].

- Inspired by survey science [45, 39], we instructed the models to **examine each option individually**. However, the models still selected too few options overall and did not improve performance (see rows 5-6 in Table 3).

Given more information, two approaches do improve performance and can provide additional insights into why the models struggle.

- **Providing the number of correct choices improves performance.** To understand how much error is due to the models' lack of knowledge regarding the number of correct options, we explicitly provided this information in the instruction for each question. This increased the exact match rate by 20.95 percentage points and reduced the selection bias metric RStd. However,when we instead provided the average number of correct choices across all questions in SATA-BENCH, performance declined (see rows 7-8 in Table 3).

- **Converting questions to multi-choice question with one correct answer.** For example, consider a question with three correct answers and six incorrect answers: we expanded it into three separate single-choice questions, each with one correct answer and six incorrect answers. We redefined the exact match rate as the percent of all original questions where a model answered all expanded questions correctly. This approach improved performance by $12.53\%$ (see row 9 in Table 3), demonstrating that SATA questions are significantly harder for LLMs than single choice questions.

Both results suggest that while models can often identify individual correct answers, they lack awareness of how many correct answers exist, which contributes to their low performance.

## 4 Improving Performance on SATA Questions

The experimental results in Section 3 demonstrate that **Selection Bias** and **Count Bias** degrade LLM performance on SATA-BENCH, and that simple prompting strategies do not lead to significant improvements. This section focuses on improving performance on open-source models, which allows us to leverage token-level logits or probability estimates from the first token prediction.

To address **Selection Bias**, we draw from prior research on token debiasing methods [9, 55] in the MCQ setting, where selection bias is attributed to the *a priori* probability mass assigned by the model to specific option IDs. These methods propose various techniques to capture and remove such biases. We hypothesize that similar debiasing techniques can be adapted to mitigate unselection bias in SATA tasks. To address **Count Bias**, we retrieve the predicted probabilities of option IDs and select options whose probabilities exceed a predefined threshold. However, because SATA-BENCH includes a large option set, the probability distribution tends to decay rapidly, with most options receiving near-zero probability mass beyond the first few choices. This makes it challenging to establish a reliable threshold. Converting SATA questions into multiple binary classification problems helps but significantly increases inference cost.

---

**Algorithm 1:** Choice Funnel

**Input** : LLM $\pi_\theta$, SATA problem $\mathcal{T}$, option set $\mathcal{O}$, $NOTA$ stop option, $\tau$ confidence threshold

\# Initialize the selected option set
$\mathcal{R} \leftarrow \emptyset$
**while** $\mathcal{O} \neq \emptyset$ **do**
    \# Generate prompt with available options
    $\mathbf{P} \leftarrow \text{MakeSATAPrompt}(\mathcal{T}, \mathcal{O})$
    \# Get first token probability distribution and apply token debiasing
    $p \leftarrow \text{DebiasingFunction}(\pi_\theta(\cdot | \mathbf{P}))$
    \# Select option with highest probability
    $o \leftarrow \arg\max_{o \in \mathcal{O}} p(o)$
    \# 1. stop when "None of the above" is selected
    **if** $o = NOTA$ **then**
        | **break**
    **end**
    $\mathcal{R} \leftarrow \mathcal{R} \cup \{o\}$
    \# 2. stop when the confidence threshold is reached
    **if** $p(o) > \tau$ **then**
        | **break**
    **end**
    **if** $length(\mathcal{R}) = 1$ **then**
        | $\mathcal{O} \leftarrow \mathcal{O} \cup \{\text{NOTA}\}$
    **end**
    $\mathcal{O} \leftarrow \mathcal{O} \setminus \{o\}$
**end**

**Output** : $\mathcal{R}$

---

**Choice Funnel Algorithm.** To improve model performance on SATA problems, we propose a decoding method called **Choice Funnel** (see Algorithm 1). This approach first adds an auxiliary option "None of the above". It then selects the option with the highest *first debiased token probability* and removes it from the option set. This process repeats iteratively until one of two stopping conditions is met: *(i) the model selects the "None of the above" option* or *(ii) the probability of the next selected option falls below a predefined confidence threshold.*

The idea of introducing "None of the above" (*NOTA*) comes from the traditional survey science domain, where options like "I don't know" (*idk*) are commonly included to improve the data collected in surveys [44]. Recent research shows that survey design principles can inform LLM development [17] and that LLMs exhibit similar biased response behaviors as humans [9, 16]. In our case *NOTA* outperforms *idk* (see ablation study in Appendix M.1).

The intuition behind the second stopping condition comes from our observation of model output probabilities, where the highest token probability tends to be lower at the beginning of iterations, since the model treats multiple options as equally correct. Later in the process, relatively higher probability is assigned to the final remaining correct option in the option set. We also show that Choice Funnel performs best when both stopping conditions are used together (see ablation study in Appendix M.3). Regarding the choice of *DebiasingFunction* in Algorithm 1, Choice Funnel is flexible and can incorporate any token debiasing method proven effective in MCQ settings. We demonstrate one such debiasing method in Section 4. Additional ablation results on each sub-component of Choice Funnel are provided in Appendix M.2. Finally, the inference cost of Choice Funnel, measured by the number of model forward passes, scales linearly with the number of *correct labels* rather than the *total number of labels*. *This makes the method particularly efficient when the correct labels represent a small fraction of the option set.*

**Experimental Setup.** We adapted the PriDe algorithm [55] as the token debiasing method in our experiments due to its label-free and computationally efficient implementation. It works by first estimating the model's prior bias towards specific option ID tokens (e.g., A, B, C) through random permutations of option contents in a small subset of test samples (10% in our experiments). We then use this estimated prior to adjust the prediction distribution on the remaining samples, separating the model's inherent positional and token biases from its task-specific predictions. Because the original PriDe algorithm was designed for standard single-answer MCQ settings, we modified it to better fit our SATA setting (see Appendix K). We evaluate the performance of Choice Funnel against three baseline methods that rely on first-token probabilities: (i) First token probability with a fixed threshold, as defined in Section 3 (referred to as *first token*). (ii) Building on method 1, we apply PriDe debiasing method [55] (referred to as *first token debiasing*). (iii) Convert each option into an individual binary yes/no question (referred to as *yes/no*). We expect *yes/no* to be a strong baseline, as it evaluates each choice independently. In this study, we use basic prompts (see Appendix H) and experiment with 7 LLMs from Table 2 that fall under the Probability Based Retrieval category (more details in Appendix L). For each model, we compute metrics reported in Table 2, and additionally report an *InfCost* metric to capture the number of model forward passes required for each method.

**Key Observations.** Choice Funnel consistently outperforms all three baselines across all 7 models in EM, SPD, and CtAcc (see Table 4). *Choice Funnel reduces unselection bias and count bias* – compared to the *first token* baseline, *Choice Funnel* achieves an average 56.16% reduction in *SPD* and 154.62% improvement in *CtAcc*, resulting in a 277.48% gain in Exact Match (EM) performance. While reasoning models also show improvements with Choice Funnel, we exclude these from aggregate calculations as their exceptionally low baselines would artificially inflate gains. When compared to our strongest baseline, the *yes/no* approach, *Choice Funnel* achieves a substantial 29.87% improvement in EM while reducing model forward passes by 64.48% thanks to its early stopping mechanism, demonstrating efficient inference scalability. Statistical significance testing (t-test) confirms that *Choice Funnel* significantly outperforms both *yes/no* and *first token debiasing* in EM and CtAcc, with a maximum p-value of 0.0079. While our models' parameter sizes (7B-14B) limit direct comparison to much larger proprietary models, Choice Funnel's performance on the *phi3-small* model still exceeds that of larger models such as Llama-90B and Mistral-Large V2 (see Table 2). This further underscores the effectiveness of our method. Additional ablation studies on the individual components of *Choice Funnel* are provided in Appendix M.

## 5    Related Work

**SATA Benchmark.** Many existing MCQ benchmarks have only one correct answer and thus cannot test LLMs' ability to select multiple correct choices. On the one hand, existing SATA datasets, such as [31, 30, 3, 27, 8], have more than 30 labels per question to choose from. This makes it impractical for LLMs to identify all correct labels from such large label pools. Other SATA-style datasets test narrow, specialized capabilities, such as emotional understanding [15] or music style understanding [54], which are less emphasized in mainstream LLM benchmarks. Since most existing methods to solve

Table 4: Performance of various models on SATA-BENCH using different decoding methods. *Choice Funnel* achieves generally better performance, effectively reducing selection and count bias compared to three baseline methods. Best values in each column are highlighted in **bold**. Columns labeled [↑] indicate higher-is-better; columns labeled [↓] indicate lower-is-better. All numeric values are rounded to two decimal places.

| Model Name | EM↑ | Precision↑ | Recall↑ | JI↑ | SPD↓ | CtDifAbs↓ | CtAcc↑ | InfCost↓ |
|---|---|---|---|---|---|---|---|---|
| Mistral-8B + *first token* | 14.73 | 81.46 | 53.23 | 46.63 | 11.42 | 1.95 | 0.21 | **1650** |
| Mistral-8B + *first token debiasing* | 8.91 | 65.17 | 37.97 | 34.27 | 152.23 | 2.34 | 0.14 | 2534 |
| Mistral-8B + *yes/no* | 16.48 | 75.49 | **55.91** | 48.80 | 12.88 | 1.94 | 0.21 | 15517 |
| Mistral-8B + *choice funnel* | **20.24** | **86.03** | 55.78 | **52.56** | **8.50** | **1.74** | **0.27** | 4803 |
| Phi3-7B + *first token* | 2.97 | **87.25** | 35.67 | 34.57 | 23.22 | 2.35 | 0.07 | **1650** |
| Phi3-7B + *first token debiasing* | 1.76 | 67.92 | 28.24 | 27.47 | 175.24 | 2.50 | 0.05 | 2534 |
| Phi3-7B + *yes/no* | 25.45 | 78.41 | **72.40** | 60.03 | **1.39** | **1.64** | 0.30 | 15517 |
| Phi3-7B + *choice funnel* | **29.27** | 83.27 | 70.24 | **61.85** | 3.47 | 1.42 | **0.38** | 6339 |
| Qwen2.5-14B + *first token* | 6.30 | **87.84** | 38.76 | 37.58 | 21.01 | 2.26 | 0.12 | **1650** |
| Qwen2.5-14B + *first token debiasing* | 4.61 | 67.95 | 31.49 | 30.36 | 154.26 | 2.43 | 0.09 | 2534 |
| Qwen2.5-14B + *yes/no* | 25.64 | 79.80 | 60.56 | 56.18 | 2.76 | 1.52 | 0.31 | 15517 |
| Qwen2.5-14B + *choice funnel* | **27.82** | 85.69 | **67.07** | **61.12** | 3.80 | **1.42** | **0.35** | 6005 |
| Bloomz-7B + *first token* | 11.27 | 66.09 | 50.80 | 41.15 | 20.62 | 1.71 | 0.20 | **1650** |
| Bloomz-7B + *first token debiasing* | 7.09 | 59.07 | 38.41 | 32.05 | 149.17 | 2.19 | 0.15 | 2534 |
| Bloomz-7B + *yes/no* | 11.93 | 39.80 | 42.67 | 29.40 | 17.78 | 3.24 | 0.13 | 15517 |
| Bloomz-7B + *choice funnel* | **20.18** | **66.62** | **54.90** | **46.15** | **9.82** | **1.71** | **0.32** | 5440 |
| Llama3-8B + *first token* | 13.82 | **80.30** | 47.37 | 43.64 | 12.09 | 1.88 | 0.22 | **1650** |
| Llama3-8B + *first token debiasing* | 7.58 | 62.83 | 32.28 | 30.38 | 151.74 | 2.34 | 0.14 | 2534 |
| Llama3-8B + *yes/no* | 14.85 | 70.30 | **65.61** | 51.43 | **1.91** | 1.78 | 0.23 | 15517 |
| Llama3-8B + *choice funnel* | **19.88** | 78.69 | 56.19 | **50.36** | 7.75 | **1.66** | **0.33** | 4975 |
| Phi4-mini-reasoning + *first token* | 2.12 | **77.98** | 30.82 | 29.69 | 21.62 | 2.39 | 0.07 | **1650** |
| Phi4-mini-reasoning + *first token debiasing* | 1.27 | 59.77 | 25.74 | 24.51 | 156.16 | 2.32 | 0.07 | 2534 |
| Phi4-mini-reasoning + *yes/no* | 4.36 | 51.08 | **81.59** | 45.24 | 7.09 | 3.19 | 0.10 | 15517 |
| Phi4-mini-reasoning + *choice funnel* | **18.42** | 74.87 | 54.84 | **49.14** | **3.30** | **1.59** | **0.27** | 6003 |
| DeepSeek-R1-Distill-Llama-8B + *first token* | 8.85 | 72.20 | 45.81 | 40.02 | 13.38 | **1.75** | 0.20 | **1650** |
| DeepSeek-R1-Distill-Llama-8B + *first token debiasing* | 5.45 | 59.29 | 31.12 | 28.48 | 134.36 | 2.14 | 0.14 | 2534 |
| DeepSeek-R1-Distill-Llama-8B + *yes/no* | 0.12 | 40.31 | **89.51** | 40.19 | 27.96 | 5.73 | 0.01 | 15517 |
| DeepSeek-R1-Distill-Llama-8B + *choice funnel* | **14.36** | **75.56** | 45.56 | **42.87** | 12.37 | 1.87 | **0.21** | 4630 |

SATA questions require converting questions to a bag-of-words [32], and as a result, most of the above datasets exist only in bag-of-words format, making them unsuitable for evaluating LLMs in our benchmark setting. To our knowledge, there is currently no existing LLM benchmark that consists exclusively of SATA questions.

**Selection Bias.** Many previous papers have discussed the tendency of LLMs to favor choices based on option order or specific symbols when answering MCQs [23, 52]. However, these papers have primarily focused on single-answer questions. A common approach to reducing selection bias involves calibrating output probabilities using the prior bias of an option ID [55]. However, it remains unclear how to define or compute such priors in SATA questions.

## 6   Conclusion

We introduced SATA-BENCH, a carefully curated suite of "Select All That Apply" (SATA) questions designed to evaluate LLMs in scenarios where multiple correct answers must be identified. Spanning diverse question types—from reading comprehension to text classification—and covering domains such as law and biomedicine, SATA-BENCH presents a comprehensive challenge for current LLMs. Our benchmarking study of both open-source and proprietary models revealed a best exact-matching accuracy of only $41.8\%$, highlighting the difficulty of selecting all the correct options. Our ablation studies indicate that a simple prompting strategy alone cannot boost exact match performance, primarily because LLMs struggle to determine the correct number of answers. This limitation stems from the fact that SATA-style questions are rarely included in LLMs' benchmark datasets, making LLMs prone to selection and count biases even when they recognize some correct options. To address this limitation, we proposed the *Choice Funnel* algorithm, which significantly improves exact match performance by systematically guiding the selection process. Our findings highlight the need for more focused research on handling multiple answer tasks, where partial correctness is insufficient. We hope that SATA-BENCH and the *choice funnel* methodology will encourage the development of more robust LLMs capable of handling realistic, multi-answer scenarios, ultimately improving their effectiveness in real-world applications that require identifying all relevant answers.

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

## Impact Statement

The introduction of SATA-BENCH marks a crucial advancement in evaluating Large Language Models (LLMs) on "Select All That Apply" (SATA) multiple-choice questions. By addressing a significant gap in existing benchmarks, which predominantly focus on single-answer multiple-choice tasks, SATA-BENCH challenges LLMs with real-world scenarios requiring multiple correct responses across domains such as reading comprehension, law, and biomedicine. This benchmark highlights the limitations of current LLMs, which struggle to accurately determine all valid answers, achieving a best-case exact match accuracy of only $41.8\%$.

SATA-BENCH's impact extends beyond evaluation, as it reveals key biases in LLM decision-making, such as count bias and selection bias, which hinder performance on multi-answer tasks. To address these shortcomings, the development of the *Choice Funnel* algorithm demonstrates a novel approach to systematically improving LLM selection accuracy, significantly enhancing model performance in SATA tasks.

While the current focus is on knowledge-intensive domains, the potential for expansion into additional fields such as mathematics, coding, and instruction following is vast. SATA-BENCH can also be extended to free-text tasks, where the set of correct responses is not explicitly provided, and to other modalities, such as voice and vision. This would further refine LLM capabilities in handling complex, multi-faceted decision-making tasks. By pushing the boundaries of LLM evaluation, SATA-BENCH lays the foundation for the next generation of AI systems capable of more nuanced, flexible reasoning in diverse real-world applications.

## Limitations

**Memorization.** While we believe most questions in SATA-BENCH have not been seen during pretraining, we cannot fully rule out the possibility that some LLMs may have been exposed, even partially, to the source datasets. We do not have access to the pretraining data of proprietary models and therefore cannot conclusively assess memorization.

**Domain Coverage.** SATA-BENCH spans six diverse domains, including reading comprehension, biomedicine, and law. However, the total number of domains remains limited compared to larger-scale benchmarks such as MMLU, indicating that further expansion is needed for broader generalization.

**Text Modality.** Our benchmark is text-only. Real-world tasks often require multimodal reasoning (e.g., interpreting charts, images, or audio), which SATA-BENCH does not evaluate. We also do not address other data modalities, such as structured tabular data or sensor data, which are common in practical applications.

**Label Correctness.** Although we perform rigorous human validation and evaluations, we acknowledge that human beings can make mistakes. Some domains, such as biomedicine and law, are inherently complex and may contain subtle ambiguities. Thus, we cannot guarantee perfect correctness of all labels despite triple human annotation and agreement filtering.

**Language Limitation.** SATA-BENCH includes only English-language questions. Evaluating multilingual capabilities or cross-lingual transfer remains a work for the future.

