# A Dataset Description

In this section, we describe the original datasets and their characteristics in detail.

**Reading Comprehension** is a dataset of short paragraphs and multi-sentence questions that can be answered from the content of the paragraph. Some questions contain multiple correct answers. The dataset we use is from (https://cogcomp.seas.upenn.edu/multirc/). The metadata is licensed under the Research and Academic Use License.

We chose this dataset for the following 3 reasons.

1. The number of correct answer-options for each question is not pre-specified. This removes the over-reliance of current approaches on answer-options and forces them to decide on the correctness of each candidate answer independently of others. In other words, unlike previous work, the task here is not to simply identify the best answer-option, but to evaluate the correctness of each answer-option individually.

2. The correct answer(s) is not required to be a span in the text.

3. The paragraphs in our dataset have diverse provenance by being extracted from 7 different domains such as news, fiction, historical text etc., and hence are expected to be more diverse in their contents as compared to single-domain datasets. The goal of this dataset is to encourage the research community to explore approaches that can do more than sophisticated lexical-level matching.

**Toxicity** is adapted from RealToxicPrompts. The dataset select prompts from sentences in the OPEN-WEBTEXT CORPUS (Gokaslan and Cohen, 2019), a large corpus of English web text scraped from outbound URLs from Reddit, for which we extract TOXICITY scores with the PERSPECTIVE API. To obtain a stratified range of prompt toxicity, we sample 25K sentences from four equal-width toxicity ranges ([0,.25], ..., [.75,1]), for a total of 100K sentences. We then split sentences in half, yielding a prompt and a continuation, both of which we also score for toxicity. For each data point, we provide the definition for each category as well as shuffle the choices for each category. We only classify the case when the category's sum of prompt and continuation score is above 1.5 for each label. The dataset we use is from (https://huggingface.co/datasets/allenai/real-toxicity-prompts). The metadata is licensed under the Apache License.

**News** is processed from Reuters text categorization test collection dataset. It contains a collection of documents that appeared on Reuters newswire. There are originally 120 related topics, where each document can be related to multiple topics. There are two challenges related to this dataset preparation: 1. The number of topics can be too large for a small number of selections. 2. Some popular topics are commonly included in the documents, making a certain choice much more popular than other choices, which can bias the models in our study. With this in mind, we limit our selection to 10 options from the 120 topics for each documents, and the remaining choices are selected randomly from the topic pool; we also re-label the choices using unique mapping per document to keep the final answers evenly distributed between all letter choices (e.g. A/B/C/D...). The dataset we use is from (https://archive.ics.uci.edu/dataset/137/reuters+21578+text+categorization+collection). This dataset is licensed under a Creative Commons Attribution 4.0 International (CC BY 4.0) license.

**Biomedicine** is adapted from the PubMed MultiLabel Text Classification Dataset, which is a collection of research articles from the PubMed repository. Originally, these documents are manually annotated by Biomedical Experts with their Medical Subject Headings (MeSH) labels, and each article are described in terms of 10-15 MeSH labels. The adopted dataset has been processed and mapped to its root level with 15 distinct MeSH labels in total. The dataset we use is from (https://www.kaggle.com/datasets/owaiskhan9654/pubmed-multilabel-text-classification). This dataset is licensed under a CC0: Public Domain license.

**Laws** is adapted from EURLEX57K which contains 57k legislative documents in English from EUR-Lex (https://eur-lex.europa.eu) with an average length of 727 words. All the documents of the dataset have been annotated by the Publications Office of EU (https://publications.europa.eu/en) with multiple concepts from EUROVOC (http://eurovoc.europa.eu/). EURLEX contains 7201 concepts. There are two challenges when converting this dataset to multi-choice question answering dataset: 1. The 7201 concepts is too big a pool for a small number of selection, most documents have <10 concepts in this dataset. 2. Some popular concepts are included in a number of documents, making a certain choice much more frequent than other choices. This is problematic because it may force the

model to learn the popular letter of choice rather than the content of the questions. With this in mind, we limit our selection to 15 options from the 7201 topics pool for each document, and the remaining choices are selected randomly from the topic pool; we also shuffle and and re-label the choices using unique mapping per document to keep the final answers evenly distributed between each letter choice. The dataset we use is from (https://paperswithcode.com/dataset/eurlex57k). This dataset is licensed under Apache License.

**Events** is adapted from the "events classification biotech" dataset, which contains diverse biotech news articles consisting of various events. The curated dataset has 3140 questions with 5 choices of events for each document. Six choices are provided for each question. The dataset we use is from (https://paperswithcode.com/dataset/events-classification-biotech). This dataset is licensed under the Open Data Commons Attribution License (ODC-By) v1.0

# B  Dataset Filtering

The Biomedicine, Law, and Events datasets were originally multi-label classification tasks, which we adapted into SATA questions by creating distractor (incorrect) choices from the unselected labels. There are two challenges when converting these datasets to SATA format: 1. Many of them have a large label pool with only a few correct answers, which is not reasonable for multiple-choice questions. 2. There can be some popular answers frequently exist in the original data, making certain choices more frequent than others. This is problematic because it may force the model to learn the popular token of choice (e.g. Choice A/a/1) rather than the content of the questions. For example, the law dataset is originally from EUR-Lex data [18] contains 57k legislative documents in English (https://eur-lex.europa.eu) annotated by the Publications Office of EU (https://publications.europa.eu/en) with over 7k concepts from EUROVOC (http://eurovoc.europa.eu/). To address the first challenge, we kept an option-to-answer ratio between 2 and 3, considering the balance between the number of correct answers and incorrect choices. The distractors were sampled randomly from the topic pool. We also shuffled and and re-labeled the choices using unique mapping per question to keep the final answers evenly distributed between each choice token. An example question from each data source is shown in Figure 4.

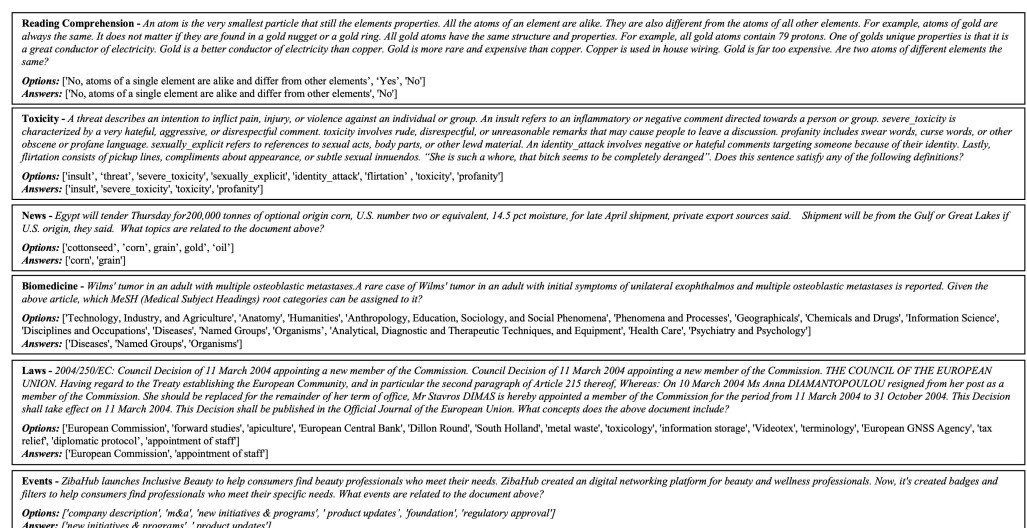

Figure 4: Representative examples of questions from various data sources used to construct SATA-BENCH.

## B.1  Initial Filtering

We manually filtered out questions that contain vague quantities, degrees of likelihood, temporal ambiguity, qualitative subjectivity, comparative uncertainty, general and undefined references. We use AWS Comprehend to remove questions that contain personal financial information or contact

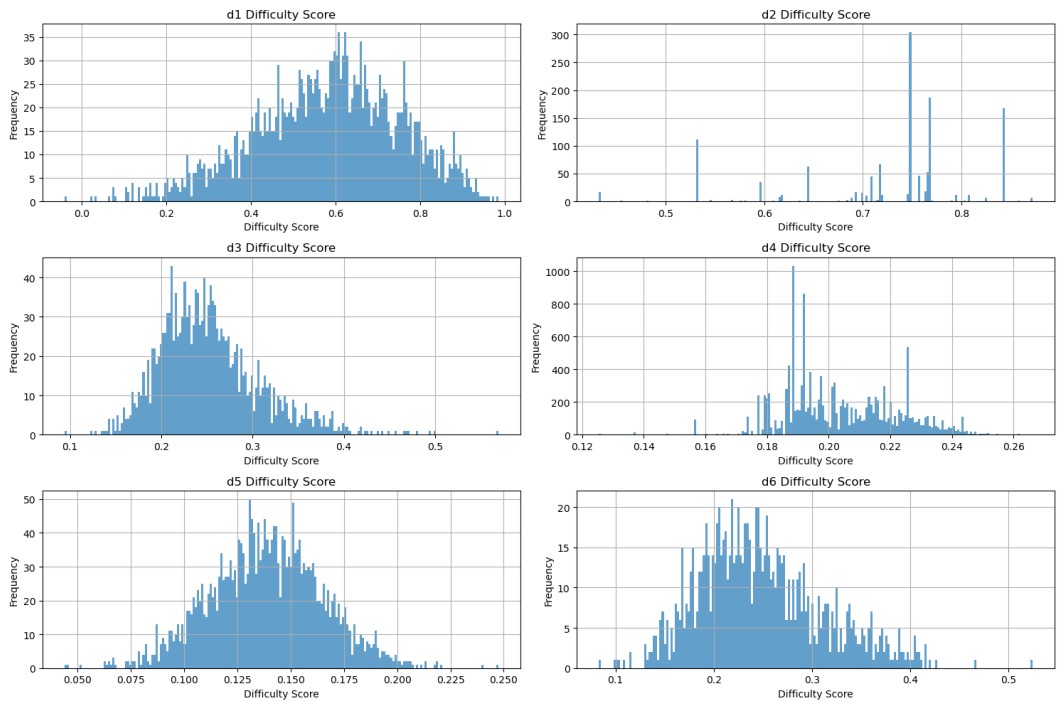

Figure 5: Confusion score distribution across all questions before filtering. d1: Reading Comprehension, d2: Toxicity, d3: News, d4: Biomedicine, d5: Laws, and d6: Events.

Table 5: Original data source statistics. We report the following metrics – n: number of instances, q: number of possible labels across the entire dataset, s: proportion of single-answer questions, m: mean number of correct answers, me: median number of correct answers, min: minimum number of correct answers, max: maximum number of correct answers, LC: label cardinality, r: ratio of the number of choices to the median number of correct answers (LC / me).

| Data Source | n | q | s | m | me | min | max | LC | r |
|---|---|---|---|---|---|---|---|---|---|
| Reading Comprehension | 5131 | na | 27% | 2.344 | 2 | 0 | 10 | 2-21 | na |
| Toxicity | 5994 | 8 | 60% | 2.639 | 2 | 2 | 7 | 8 | 4 |
| News | 11360 | 120 | 83% | 2.567 | 2 | 2 | 16 | 6 | 3 |
| Biomedicine | 50000 | 15 | 0.07% | 5.745 | 6 | 0 | 13 | 15 | 2.5 |
| Laws | 57000 | 7201 | 0.54% | 5.069 | 5 | 1 | 26 | 15 | 3 |
| Events | 3140 | 29 | 50.7% | 2.683 | 2 | 2 | 5 | 6 | 3 |

information. We leave questions that contain public available information such as the company name and address. All filtered words are mentioned below in Table 6.

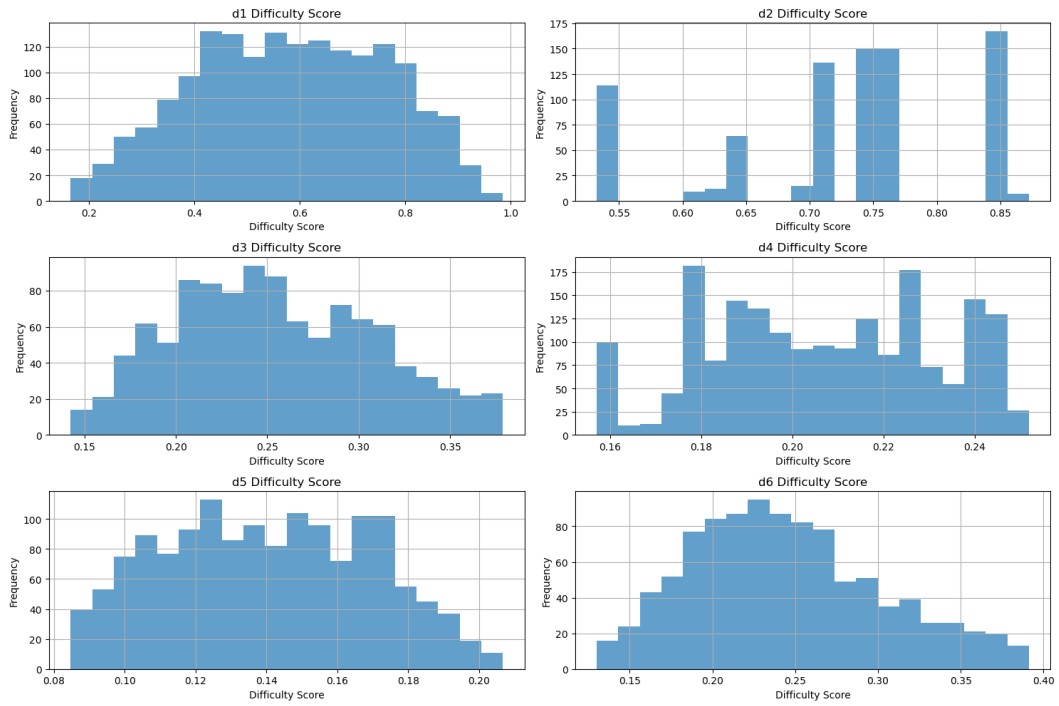

Figure 6: Confusion Score distribution of the filtered questions. d1: Reading Comprehension, d2: Toxicity, d3: News, d4: Biomedicine, d5: Laws, and d6: Events.

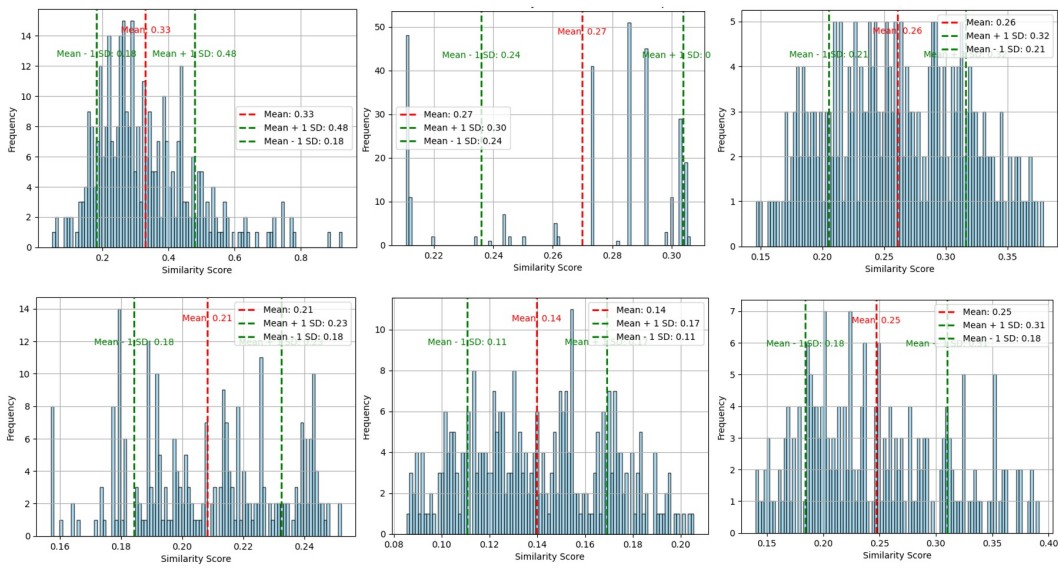

Figure 7: Confusion Score distribution separately visualized for each source dataset. (Left to right) Top row: Reading Comprehension, Toxicity, News; Bottom row: Biomedicine, Laws, Events.

Table 6: Identified categories of vague terms along with representative examples

| Category | Examples |
|---|---|
| Vague Quantities | some, several, many, few, a lot, plenty, numerous, various, partially, a handful, a bit, a portion |
| Degrees of Likelihood | maybe, possibly, probably, likely, unlikely, apparently, presumably, seemingly, conceivably, arguably, occasionally |
| Temporal Ambiguity | sometimes, often, rarely, occasionally, once in a while, from time to time, now and then, every so often |
| Qualitative Subjectivity | bad, nice, significant, substantial, important, interesting, sufficient, adequate, reasonable, moderate |
| Comparative Uncertainty | more or less, about, around, roughly, close to, kind of, sort of, nearly, almost, approximately |
| General and Undefined References | thing, things, anything, everything, whatever, such, kind, type, sort |

## B.2 Human Validation

Human validation is to ensure that the questions are unambiguous. Using humans to validate the question is inspired by [49, 33]. For each question in the benchmark, we ask five annotators whether the question contains ambiguous information.

---

**Human Validation**

You are presented with the following:
Paragraph: *paragraph*
Question: *question*
Choices: *choice*
The question text and answer choices are clearly written:
*Strongly agree*
*Agree*
*Neither agree nor disagree*
*Disagree*
*Strongly Disagree*
Answers:

---

Once it is done, the total cost is tracked (1301.89), with 5 people per label at a cost of 0.012 each. We only select questions that are "Strongly agree" and "Agree" > 0.8.

## B.3 Human Labeling

To ensure that each question has a valid and correct answer, we conducted a comprehensive human evaluation. An initial manual inspection revealed that some questions lacked clearly correct answers. To verify answer correctness, we recruited three experienced annotators to review all 1,650 questions that remained after prior filtering and validation. Annotators were compensated at a rate of at least $35 per hour. Each question was independently evaluated by at least two annotators.

For each question, the original reference answer and four anonymized LLM-generated answers (from Claude 3.7, GPT-4 Turbo (O3), Grok 3, and Gemini 2.5) were provided. In cases where the two annotators disagreed, a third annotator reviewed the original answer, all LLM answers, and both annotators' decisions to determine the final label or to discard the question. Detailed annotation guidelines were provided below. As a result of this process, 47 questions were discarded due to ambiguity or disagreement, and an additional 46 were removed for quality-related issues.

Given original answers and LLMs' answers, you'll try to identify correct answer of the following questions. You're expected/encouraged to use Google, and any internet resources you can find to try and answer the question correctly.

**Requirements and Expectations** 1. You are encouraged to use Google, and any websites you can think of or find that may help you answer the question and understand the concept. However, you are NOT allowed to use AI assistants like chatGPT, Claude, Grok3 Geimini, etc., or ask people for help. All their answers to the question has been provided anoymously under LLM Answers.

2. We ask that you spend at least 5 minutes trying to answer each question before making your selection. If you haven't settled on an answer choice in that time, we encourage you to spend as long as you need to be confident in your selection.

3. These questions will be hard, and you will likely need to spend a while on each of them to make progress understanding the context. Read relevant resources, take plenty of time, and answer "I don't know" if you're pretty sure you have no realistic way of answering confidently.

4. You will also be given the opportunity to give feedback on the question. We're especially interested in feedback about whether the question was ambiguous, but please feel free to give feedback of any other form!

**Suggestions and Strategies for Labeling** 1. Look up definitions for all of the unfamiliar terms in the question and answer choices. Keep a list of those definitions handy so you can easily refer back to the definitions if you forget the jargon.

2. LLMs' answer is not always reliable and original answer is not always correct. Please try to solve the question independently before looking at potential answers.

2. Look for primary resources, like research papers and textbooks, as these can often contain clearer explanations than sources like Wikipedia (although Wikipedia can be useful in many cases as well).

You are presented with the following:
Paragraph: *paragraph*
Question: *question*
Choices: *choice*
Original Answers: *original answer*
LLM Answers: *llm answers*
Answers:

## C Hyperparameters

To ensure consistent and high-quality outputs across different models, we standardized the decoding hyperparameters for most model generations by setting the temperature to 0 (to promote deterministic outputs), top-p (nucleus sampling) to 0.95 (to allow for a balance between diversity and relevance), and a maximum token limit of 1,024 tokens. Recognizing the enhanced reasoning capabilities of certain models, we adjusted the configurations accordingly. For O3 and Grok 3, we set the thinking budget to be high. For Geimini 2.5 thinking and Claude 3.7 Thinking, we set the thinking budget to be 16k. For R1, we set max tokens 16k. This is to provide enough budget for reasoning models to finish thinking.

## D Compute Resources

We use AWS Bedrock batch inference for large models' inference such as Claude3 Sonnet, Claude 3.5 Haiku, Claude 3 Haiku, Claude 3 Opus, Nova Pro, Nova Lite, Claude 3.5 Sonnet, Llama 3.1 405B, Mistral Large V2, Nova Micro, Llama 3.2 90B, and Llama 3.1 70B. We use AWS cross-region inference for Claude3.7 Reason, Claude3.7, and Deepseek R1. We use official APIs from the respective providers for models such as OpenAI O3, GPT4.1, Grok3 Reason, GPT4, Geimini2.5 Reason, Gemini 2 Flash, GPT 4.1 mini, and Qwen Plus.

For experiments that require accessing model's hidden states and log probs. We run inference on one EC2 $p4d.24xlarge$ (Nvidia A100 40GiB GPU) instance and one EC2 $g5.4xlarge$ (Nvidia A10G

Table 7: Model cards summarizing specifications and details for all evaluated large language models.

| Model Name | Creator | Complete Model ID | Release | Hosting |
|---|---|---|---|---|
| O3 | OpenAI | o3-2025-04-16 | 04/16/25 | OpenAI API |
| GPT-4.1 | OpenAI | gpt-4.1-2025-04-14 | 04/14/25 | OpenAI API |
| Grok 3 Think | xAI | grok-3-mini-beta | 02/19/25 | xAI API |
| GPT-4-turbo | OpenAI | gpt-4o-2024-11-20 | 11/20/24 | OpenAI API |
| Claude-3.7 Sonnet Think | Anthropic | anthropic.claude-3-7-sonnet-thinking-20250219-v1:0 | 02/24/25 | AWS Bedrock |
| Claude-3.7 Sonnet | Anthropic | anthropic.claude-3-7-sonnet-20250219-v1:0 | 02/24/25 | AWS Bedrock |
| Claude-3 Sonnet | Anthropic | anthropic.claude-3-sonnet-20240229-v1:0 | 02/29/24 | AWS Bedrock |
| Gemini 2.5 Think | Google | gemini-2.5-pro-preview-03-25 | 03/25/25 | Vertex AI |
| Claude-3.5 Haiku | Anthropic | anthropic.claude-3-5-haiku-20241022-v1:0 | 10/22/24 | AWS Bedrock |
| Claude-3 Haiku | Anthropic | anthropic.claude-3-haiku-20240307-v1:0 | 03/07/24 | AWS Bedrock |
| Claude-3 Opus | Anthropic | anthropic.claude-3-opus-20240229-v1:0 | 02/29/24 | AWS Bedrock |
| Gemini 2 Flash | Google | gemini-2.0-flash | 02/05/24 | Vertex AI |
| GPT-4.1 mini | OpenAI | gpt-4.1-mini-2025-04-14 | 04/14/25 | OpenAI API |
| Nova Pro | Amazon | amazon.nova-pro-v1:0 | 12/03/24 | AWS Bedrock |
| Claude-3.5 Sonnet | Anthropic | anthropic.claude-3-5-sonnet-20240620-v1:0 | 06/20/24 | AWS Bedrock |
| Llama 3.1 405B | Meta | meta.llama3-1-405b-instruct-v1:0 | 07/23/24 | AWS Bedrock |
| Nova Lite | Amazon | amazon.nova-lite-v1:0 | 12/02/24 | AWS Bedrock |
| DeepSeek R1 | DeepSeek | deepseek.r1-v1:0 | 01/20/25 | AWS Bedrock |
| Mistral Large V2 | Mistral AI | mistral.mistral-large-2407-v1:0 | 07/24/24 | AWS Bedrock |
| Qwen Plus | Alibaba | qwen-plus-2025-04-28 | 04/28/25 | Alibaba API |
| Nova Micro | Amazon | amazon.nova-micro-v1:0 | 12/02/24 | AWS Bedrock |
| Llama 3.2 90B | Meta | meta.llama3-2-90b-instruct-v1:0 | 09/25/24 | AWS Bedrock |
| Llama 3.1 70B | Meta | meta.llama3-1-70b-instruct-v1:0 | 07/23/24 | AWS Bedrock |
| Mistral 8B Instruct | Mistral AI | mistralai/Mistral-8B-Instruct-2410 | 10/09/24 | Hugging Face |
| Llama 3 8B | Meta | meta-llama/Llama-3.1-8B-Instruct | 07/23/24 | Hugging Face |
| BLOOMZ 7B | BigScience | bigscience/bloomz-7b1 | 07/11/22 | Hugging Face |
| DeepSeek R1 Distill 8B | DeepSeek | deepseek-ai/DeepSeek-R1-Distill-Llama-8B | 02/01/25 | Hugging Face |
| Qwen 2.5 14B | Alibaba | Qwen/Qwen2.5-14B | 09/19/24 | Hugging Face |
| Phi-3 7B | Microsoft | microsoft/phi-3-small-128k-instruct | 05/21/24 | Hugging Face |
| Phi-4-mini-reasoning | Microsoft | microsoft/phi-4-mini-reasoning | 04/15/25 | Hugging Face |

24GiB GPU) in Sydney(ap-southeast-2) region. We have also attached 8000GiB disk volume with AL2023 Linux OS image. We use HuggingFace and PyTorch as the main software frameworks.

## E    Non-expert Human Benchmark

To contextualise LLM results on SATA-BENCH, we recruited non-expert annotators on *Amazon Mechanical Turk*, adapting the instructions from [43]. All 1604 questions was labelled as follows:

- **Task set-up.** Each question was presented with the original answer options *plus decoys* (e.g. ABCD→ ABCDEFGHIJK) to identify inattentive workers. Nine independent annotations were collected per item at a rate of *$0.84 per question*, matching the fair-wage recommendations of GPQA.

- **Quality safeguards.** Workers were: (i) informed that every item contains *at least two* correct answers; (ii) forbidden from consulting LLMs or other people, yet allowed to look up unfamiliar terms on Google/Wikipedia; (iii) required to spend $\geq 2$ *minutes* on each question. Submissions that selected any decoy, took $< 1$ min, or violated the lookup policy were discarded (7.1 %).

- **Label selection.** From the surviving pool, we randomly drew one annotation as the *human label*; single-choice answers were retained to keep the evaluation comparable to LLMs that sometimes return only one option.

| | EM | Precision | Recall | JI | RStd | RSD | SPD | CtDif | CtAcc | CtDifAbs |
|---|---|---|---|---|---|---|---|---|---|---|
| Human | 17.9 | 60.6 | 54.4 | 45.0 | 15.3 | 0.46 | 1.46 | −0.6 | 34.1 | 1.44 |

Table 8: Aggregate performance of crowd annotators on the SATA-Bench subset.

As anticipated, non-experts achieve modest exact-match and precision, yet their selection-bias metrics (RStd, RSD, SPD) resemble those of mid-tier LLMs. Crucially, they exhibit *smaller absolute count bias* (|CtDif|) and higher correct-count accuracy (CTACC), indicating superior intuition for the number of correct options even when individual labels are missed. These human baselines therefore offer a realistic point of comparison for evaluating LLM performance on specialised SATA tasks.

### E.1 Non-expert Human Benchmark Instructions

We have provided details on human benchmark instructions.

---

**Human Benchmark Instructions**

You will see a short **Paragraph**, a **Question**, and a list of answer options labelled `A B C D E F G H I J K L M N O`. Your task is to mark *all* choices that you believe are correct.

**Requirements and Expectations**

1. **External resources.** You may consult Google, Wikipedia, journals, textbooks, or any other online materials that help you understand the content. **Do not use AI assistants** (ChatGPT, Claude, Gemini, Grok, etc.) and do not ask other people.

2. **Effort.** Spend **at least 2 minutes** on each item before submitting. If you still feel unsure, keep researching until you are confident, or choose "*I don't know*" if you cannot answer reliably.

3. **Difficulty.** Many items are specialised and may require careful reading. Take your time; thorough work is valued more than speed.

4. **Feedback.** After answering, you may leave comments (e.g. ambiguity, unclear wording). Constructive feedback is highly appreciated.

**Suggestions and Strategies**

1. Look up definitions of every unfamiliar term in the paragraph, question, and answer options. Keep your notes open for quick reference.

2. Approach the question *independently*—do not try to guess a "majority" answer. Rely on primary sources (research articles, textbooks) whenever possible.

3. Remember that there are *at least two* correct letters, but possibly more. Select every option you deem correct.

**Fields Presented to You**

**Paragraph:** *{{paragraph}}*

**Question:** *{{question}}*

**Choices:** *{{A. . . O}}*

**Your Answers (mark all that apply):**

_______________________________________________

**Optional Feedback:** _______________________________________

---

## F  Metrics Definition

**Performance.** The standard SATA performance metrics [48] are Exact Match (EM), Jaccard Index (JI), Mean Average Precision (Precision), and Mean Average Recall (Recall). EM captures a model's ability to select all correct answers without error, indicating its completeness in prediction. JI measures the fraction of predicted labels that match the ground-truth labels.

**Selection Bias.** Selection bias is a model's preference for selecting specific option IDs as answers, and is measured by RStd [56] and RSD [11, 42]. We also observe that some models prefer to avoid selecting certain options, which we call **unselection bias**. We introduce a metric called *Selection Probability Divergence* (SPD) to measure this type of bias. Appendix G gives more details on this issue and the design of SPD.

**Count Bias.** We observe that models tend to select fewer number of options compared to the ground truth. We refer to this as *Count Bias*. To evaluate the severity of this type of bias, we measure the following: (i) Mean difference between the number of selected options minus correct options (CtDif), (ii) Mean absolute difference between the number of selected options minus correct options

(CtDifAbs), and (iii) Percentage of cases where the model selects exactly the correct number of options (CtAcc).

## F.1  Performance Metrics Definition

Here are some standard metrics used in the literature to track performance on SATA questions.

- **Exact Match** counts how many times the entire set of predicted labels for a sample exactly matches the entire set of ground truth labels. It is then divided by the total number of samples. A perfect exact match score (1.0) means the model got every instance's labels exactly correct.

- **Jaccard Index** calculates the fraction of predicted labels that exactly match the ground truth labels—or put differently, divide the size of the intersection of predicted and true labels by the size of the union of predicted and true labels, and then average this ratio across all instances for the final score. This metric treats each label decision independently and is a good measure when we care about partial correctness in multi-label settings.

- **Recall** looks at how many labels were correctly predicted (intersection) out of how many total true labels exist. Then it averages this fraction across all instances.

- **Precision** calculates how many labels were correctly predicted (intersection) out of all the labels the model predicted. Then it averages this fraction across all instances.

## F.2  Selection Bias Metrics Definition

Here are some standard metrics to track SATA questions selection bias. These metrics are extension of existing selection bias literature.

- **Standard Deviation of Recalls (RStd)** is the standard deviation of the class-wise recall:

$$\text{RStd} = \sqrt{\frac{1}{k} \sum_{i=1}^{k} (r_i - \bar{r})^2}, \tag{1}$$

  where $k$ is the number of choices, $r_i$ is the recall of the $i$-th class, and $\bar{r}$ is the arithmetic mean of $r_i$ values. Note that our recalls are calculated at the label level since this is multi-class question [56]

- **Relative Standard Deviation (RSD)** is the class-wise accuracy standard deviation normalized by the overall accuracy:

$$\text{RSD} = \frac{\sqrt{\frac{1}{k} \sum_{i=1}^{k} (s_i - \bar{s})^2}}{\bar{s}}, \tag{2}$$

  where $k$ is the number of choices, $s_i$ is the accuracy of the $i$-th class, and $\bar{s}$ is the mean accuracy averaged across classes. Please note that our recalls are calculated at the label level since this is multi-class questions [11, 42]

## F.3  Count Bias Metrics Definition

- **CtDif** calculates the average difference in count between predicted and actual selected options. A positive value indicates that the predictions tend to select more options than the actual answers, while a negative value suggests the opposite.

- **CtDifAbs** calculates the absolute value of the average difference in count between predicted and actual selected options. A larger value indicates that the predictions tend to select the number of options that are different from the correct number of options.

- **CtAcc** calculates the proportion of predictions that select the exact same options as the ground truth labels. It provides a measure of how often the model selects the same number of answers as the true answer set.

### F.4 Additional Metrics Definition

- **InfCost** measures the number of model forward passes used for a method to complete the benchmark. A larger value indicates that the method requires more compute FLOPs and is thus more expensive. A small value indicates the method requires fewer compute FLOPs and is thus more cost-effective.

## G Unselection bias metric

We view a SATA problem as multiple binary selection problems, where each option is examined independently to be selected or passed. In our experiments, we have observed that LLMs tend not to select (i.e., skip) certain labels more frequently than others. To quantify this non-selection bias, we define a metric below, named selection probability divergence (SPD), to measure the misalignment between the ground truth and the LLM's prediction.

$$\text{SPD} = \sum_{i=1}^{k} \left(1 - \frac{q_i}{p_i}\right) \ln \frac{p_i}{q_i}, \tag{3}$$

where $k$ is the number of choices, $p_i$ is the ground truth probability of label $i$ being one of the correct choices, and $q_i$ is the prediction probability of label $i$ being one of the selected choices.

SPD has a minimal value of 0 at $q_i = p_i$ for all $i$, when the prediction aligns with the ground truth. SPD diverges as $q_i \to 0$ while $p_i$ is finite for any $i$, when the LLM shows a non-selection bias against a particular label. SPD also diverges as $p_i \to 0$ while $q_i$ is finite for any $i$, when the LLM shows a selection bias toward a particular label. In this sense, SPD serves as a metric to measure the disagreement of choice probability between the ground truth and the prediction, reflecting both under-selection and over-selection. (See Appendix G.2 for the mathematical analysis.)

### G.1 Behavior of SPD Metric

We conduct a numerical experiment to compute SPD with varying $p_i$ and $q_i$. We set the number of choices to 4, and use a Boolean list of size 4 to indicate which options are correct. Eg. for choices A, B, C, and D, the list [True, False, True, True] means the answer to the SATA question is ACD.

For the ground truth list, we sample each element of the Boolean list with a ground truth probability, p. For the prediction list, we sample the first element of the Boolean list with a prediction probability, q, and sample the other elements with probability p. With this setting, we focus on the misalignment between the ground truth and the prediction in a single label (the first label in this case).

We repeat the above sampling process $M$ times, and compute the True rate of each option for the ground truth $p_i$ and the prediction $q_i$, with $i = 1, 2, 3, 4$. We then substitute the numbers into Eq. (3) to calculate SPD. Note that in the current setting, $p_i = \text{p}, \forall i$, and $q_1 = \text{q}$, $q_{2,3,4} = \text{p}$.

Figure 8 shows the SPD-q curves under different values of the ground truth probability p. Each curve is obtained by averaging over 100 replicates, and the shaded area shows the standard deviation. The minimal value of SPD is 0 and occurs at $\text{q} = \text{p}$.

### G.2 Sensitivity of SPD to label probability ratio

We analyze the behavior of SPD as the relationship between $p_i$ and $q_i$ changes. We first define the ratio of the two probabilities as $r_i \equiv q_i/p_i$, $i = 1, 2, \ldots, k$, and rewrite the SPD definition Eq. (3) as

$$\text{SPD} = \sum_{i=1}^{k} (1 - r_i) \ln \frac{1}{r_i}. \tag{4}$$

As the misalignment between the ground truth and the prediction grows, either with $r_i \to 0$ or $r_i \to +\infty$, SPD diverges according to Eq. (4). Therefore, a large value of SPD reflects the disagreement of the choice probability between the ground truth and the prediction.

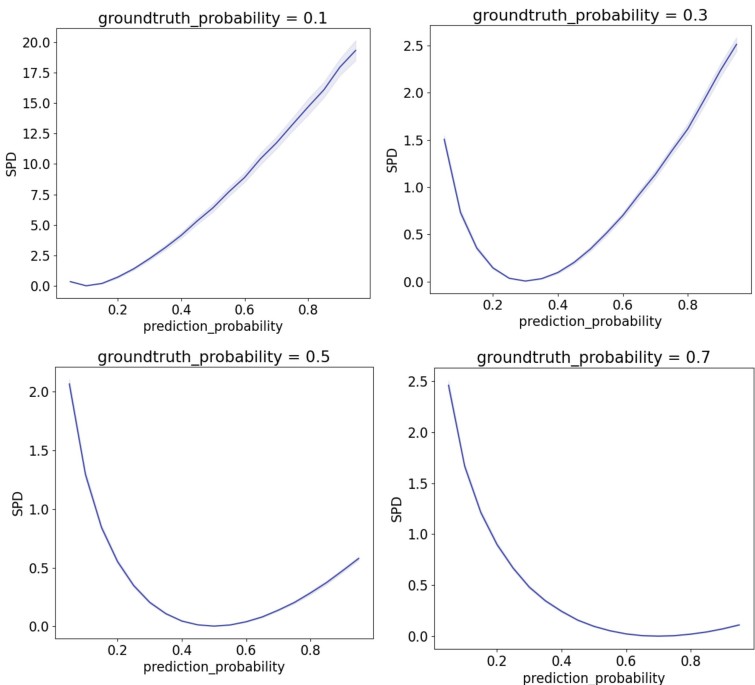

Figure 8: Relationship between Selection Probability Divergence (SPD) and prediction probability (q) across different ground truth probabilities (p). The curves are averaged over 100 replicates, and the shaded area represents the standard deviation. In each plot, the minimal value of SPD is 0 at q = p, when the prediction aligns with the ground truth.

To find the minimum of SPD, we take the partial derivative with respect to each variable $r_i$, and set it to be 0. Then we have the equations below.

$$\frac{\partial \mathrm{SPD}}{\partial r_i} = \ln r_i + \frac{r_i - 1}{r_i} = 0, \quad i = 1, 2, \ldots, k. \tag{5}$$

This set of equations has only one real solution:

$$r_i = 1, \quad i = 0, 1, \ldots, k. \tag{6}$$

Thus the SPD is minimized when $q_i = p_i$, *i.e.* when the prediction probability matches the ground truth probability for each option and when there is no bias toward or against any choice. The minimal value of SPD is 0.

# H   Prompts used in experimentation

## H.1   Prompts for open-source models

We designed simple, basic prompts without elaborate prompt engineering for all experiments with open-source models in Section 3. The main reason is that we want to avoid potential biases introduced by complex prompt engineering, thereby emphasizing the evaluation of the method itself.

### H.1.1   Choice funnel Prompt

This prompt is used for *Choice Funnel* as well as two baseline methods: *first token* and *first token debiasing*

> **Open Source Prompts**
>
> You are presented with the following:
> Paragraph: *paragraph*
> Question: *question*
> Choices:
> *Option A*
> *Option B*
> *Option C*
> *Option D*
> *Option E*
> Task:
> Identify and select all the correct answers based on the paragraph and the question.
> Answers:

### H.1.2 Yes/No for open-sourced models

This prompt is used for *yes/no* baseline method to compare against *Choice Funnel*.

> **Yes/No Prompts**
>
> You are presented with the following:
> Paragraph: *paragraph*
> Question: *question*
> Statement: *Option A |B |C |D |E*
> Task:
> Determine if the statement answers the question correctly and reply with "Yes" or "No" only.
> Answer:

## H.2 Prompts for proprietry model

### H.2.1 Problems for current MCQ prompts

Existing benchmarks [25] use the following prompts for MCQ questions and then use exact match to get the correct option.

> **Example MCQ Prompt 1**
>
> What is the correct answer to this question: *question*
> Choices: *choices*.
> Let's think step by step:
> Based on the above, what is the single, most likely answer choice?
> Answer in the format:
> correct answer is (insert answer here).

> **Example MCQ Prompt 2**
>
> Given the following question and four candidate answers (A, B, C, and D), choose the best answer.
> Question: *question*
> Your response should end with:
> The best answer is [the_answer_letter]
> where the [the_answer_letter] is one of A, B, C, or D.

However, it does not work well when there are multiple answers as LLMs are likely to repeat the content of the option before output the second choice. As an example, the below answer is A. Wheat, C. Grain instead of AC, which makes it hard to parse.

> **Example MCQ Prompt Example 1**
>
> What is the correct answer to this question: Bangladesh passed on its weekend tender for 100,000 tones of optional origin soft wheat, private export sources said. What topics are related to the document above?
> Choices: A. Wheat, B. Barley, C. Grain, D. Retail, E. Money-supply, F. Sugar]. Let's think step by step:
> Based on the above, what are all most likely answer choice?
> Answer in the format:
> correct answer is (insert answer here).
> Answer (from GPT4o): Let's analyze step by step:
> The document mentions wheat: The phrase "100,000 tonnes of optional origin soft wheat" directly refers to wheat.
> Relevant choice: A. wheat. The document does not mention barley: There is no reference to barley anywhere in the text.
> Not a relevant choice: B. barley. Grain is a broader category that includes wheat: Since wheat is a type of grain, it makes sense to include grain as a related topic.
> Relevant choice: C. grain. The document does not mention retail: The context of the tender does not relate to retail or consumer sales.
> Not a relevant choice: D. retail. The document does not mention money supply: There is no financial or monetary context provided.
> Not a relevant choice: E. money-supply. The document does not mention sugar: There is no reference to sugar in the text.
> Not a relevant choice: F. sugar. Final Answer:
> Correct answer is **A. Wheat, C. Grain**.

> **Example MCQ Prompt Example 2**
>
> What is the correct answer to this question: Bangladesh passed on its weekend tender for 100,000 tones of optional origin soft wheat, private export sources said. What topics are related to the document above? Choices: A. Wheat, B. Barley, C. Grain, D. Retail, E. Money-supply, F. Sugar. Let's think step by step:
> Based on the above, what are most likely answer choices?
> Answer in the format:
> correct answer is (insert answer here).
> Answer (from GPT4o): Let's analyze the question step by step:
> Key Subject of the Document: The document discusses Bangladesh passing on its tender for 100,000 tonnes of optional origin soft wheat.
> Relevant Topics:
> The keyword wheat clearly relates to the document. Wheat falls under the broader category of grain, making it a relevant choice. Other terms like barley, retail, money-supply, and sugar are not directly related to the subject of the document. Most Likely Answer Choices:
> A. Wheat: Directly mentioned in the document. C. Grain: Relevant as wheat is a type of grain. Correct Answer:
> The correct answer is **A Wheat, C Grain**.

We can observe that parsing the above prompts is much harder because we do not know when they will output the option ID plus answers, and we cannot blindly extract all matched option IDs. Following [25] page 38, we used the JSON format to extract answers. Closed-source LLMs have shown good instruction following capability for JSON format output [57]. Also, python has many packages that can do fuzzy JSON matching, which can be used to handle edge cases. We extract as many JSON files as possible from each output and combine the extracted answers together as the final answer to the question.

One can observe that our proposed prompts can easily extract the answer because they contain only the option IDs.

## H.3 Ablation Prompts

### H.3.1 Few Shot prompt

We report few few-shot prompt where the number of examples is equal to 5.

## H.4  Think Option by Option prompt

Inspired by [46, 39], we instruct LLM to understand each options and analyze each answer independently.

### H.4.1  Few Shot Option prompt

We further provide a few examples to teach LLMs how to think option by option, but it still does not improve the performance.

### H.4.2 Prompt with Average Options Count

### H.4.3 Prompt with Correct Number of Options

### H.4.4 Single Choice Prompt

To ensure consistency, we use a similar prompt for single choice. We use the same method to retrieve the correct choices. If there is more than one correct choice, we randomly sample from among them.

> **Single Choice Prompt**
>
> Given the following question where there is only one correct answers, choose the correct answer.
> Question: *question*
> Choices: *choices*
> Please the correct choice that apply.
> Let's think step by step: You must present your selected option IDs in the following JSON format: $\{"choice" :< A|B|C|D|E|F|G|H|I|J|K|L|M|N|O >\}$

## H.5 Prompt with Numeric Option

For numeric options, it is hard to retrieve since the number of options can be above 10, and the previous retrieving method could retrieve 12 as 1 and 2. We instruct LLMs to produce correct answers in ascending order. We start by retrieving a larger number that is above 10. For each successful retrieval, remove that number from the output. This way, we can avoid the above scenario.

> **Numeric Prompt**
>
> Given the following question where there is more than one correct answer, choose all correct answers.
> Question: *question*
> Choices: *choices*
> Please select all choices that apply. You must focus on the question and select all choices that apply. You must present your answers in ascending orders. Let's think step by step: You must present your selected option IDs in the following JSON format: $\{"choices" :< 1|2|3|4|5|6|7|8|9|10|11|12|13|14|15 >\}$

## H.6 Prompt with small alphabet Option

> **Small Alphabet Prompt**
>
> Given the following question where there is more than one correct answer, choose all correct answers.
> Question: *question*
> Choices: *choices*
> Please select all choices that apply. You must focus on the question and select all choices that apply. Let's think step by step: You must present your selected option IDs in the following JSON format: $\{"choices" :< a|b|c|d|e|f|g|h|i|j|k|l|m|n|o >\}$

# I Inference Error Handling

For $2.897\%$ of all cases, we cannot find any match in JSON format, so we use Claude 3 Haiku to extract the final labels. To be specific, we adopt the following system prompt:

**Edge Case Handling Prompt**

Given the following text, please identify **all** valid choices. A valid choice is any single letter from A to Q, which might appear right after a colon (e.g., `choices: "B"`).

- If one or more valid choices are found, concatenate them and return them in the format `<answer></answer>`: For example, `<answer>BEM</answer>`.

- If no valid choices are found, return `<answer></answer>`.

String to analyze: `<output>`

For all cases below, our Claude 3 haiku is able to accurately produce the correct outcome.

Table 9: Comparison of raw LLM outputs and the extracted labeled results obtained using Claude 3 Haiku.

| LLM Output | Claude 3 Haiku Extraction |
|---|---|
| I can't fulfill that request. | NaN |
| `"choices": { "choice": "B" }` | B |
| `{{ "choice": <B \| E \| H \| J \| L \| M \| O> }} }}`'''json | BEHJLMO |
| `{ "choice": [] }` ''' | NaN |

We then use Amazon Groundtruth labeling to check whether Claude 3 Haiku correctly parses the answer. Of those, only 47 cases were labeled as No or Yes with confidence lower than 0.6. We manually investigated those 47 cases and found that only four were actually incorrect.

Table 10: Examples of LLM outputs and corresponding extraction results where Claude 3 Haiku produced incorrect extractions.

| LLM Output | Claude 3 Haiku Extraction | Human Corrected Answers |
|---|---|---|
| Let's analyze the text and MeSH categories step by step:...: your selected option IDs - C (Organisms), your selected option IDs - E (Phenomena and Processes), your selected option IDs - G (Chemicals and Drugs) | CE | CEG |
| `{{ "choice": <D \| E \| K \| L \| M> }} }}`'''json | DELM | DEKLM |
| `{ "choice": "choice": "N"oneyour se-lected option IDs } ` ''' | N | NaN |
| Let's analyze the document step by step: ... your selected option IDs your selected option IDs. Based on this analysis, the applicable choices are A, B, C, and E. | ABC | ABCE |

## J More Details on Key Observations

**Unselection Bias.** FP/FN means False Positive Count divided by False Negative Count. If a model has 100 False Negative cases of A, it means that the model has not predicted A in 100 cases where it should have predicted A. If a model has 20 False Positive cases of A, it means that the model has predicted A in 20 cases where it should not have. The low FP/FN rate means that out of all cases, the model tends not to predict A instead of overpredicting A. Due to Count Bias, most of the models have FP/FN rate below 1. However, almost all models has one label with an extremely low FP/FN rate. For example, Nova-Micro has a label A FP/FN rate equal to 0.05 while its second worst is 0.12.

Similarly, Claude3-Haiku has a label A FP/FN rate equal to 0.27 while its second worst is 0.48 as shown in Figure 10.

Recall Difference is another metric to demonstrate unselection bias. Low recall on certain label means that LLMs' incapability of predicting certain labels correctly. As shown in Figure 9, there are many models whose worst label is more than $5\%$ below their average performance.

**Count Bias.** Figures 11 shows that nearly all models select too few responses and that this tendency increases as the number of correct answers increases. Figure 12 demonstrates that EM also falls as the number of correct answers increases. This demonstrates that LLMs tend to underpredict the number of correct choices.

# K    PriDe Debiasing Algorithm Adaptation for SATA

## K.1    PriDe Introduction

The original PriDe algorithm [56] is designed for processing MCQ question sets with fixed option set length (usually 4). It works by observing the probability changes when performing permutations of option IDs for each question, and it can compute *priors*, which is known as the probabilistic mass that the model a priori assigns to option ID tokens.

Here is an example to better illustrate the process:
Given a question set with 4 options, we compute the prior of each question from 10% of the data, take the average on each option ID position and then we get:
$$P(prior) = [0.4, 0.2, 0.2, 0.2]$$

The list corresponds to probabilities for ABCD. In this case we can see that the model biases towards option "A". Now given a new question with probabilities computed as:
$$P(observed) = [0.5, 0.3, 0.1, 0.1]$$

Without debiasing model will select option "A" as top answer. We need to subtract prior:
$$P(debiased) = P(observed) / P(prior)$$
$$P(debiased) = [1.25, 1.5, 1.0, 0.5]$$

Option "B" becomes top-1 after we remove the heavy prior on "A". To learn more low-level details, please refer to the original paper [56]).

## K.2    Limitation of Original Algorithm.

However, the prior is computed on a fixed length of 4, so the prior computed for each option has its own probability distribution. For a dataset with variable lengths of option sets (3-15 options for our SATA-Bench). We can only use priors computed for their own length groups (for example, using a length-3 prior to remove bias only for questions that have 3 options). Therefore, we might not have enough data to build an accurate prior. For example, SATA-BENCH contains only 52 out of 1650 questions with 3 choices.

**Adaptation to solve SATA questions.** To solve the above problem, we first construct a dictionary with key as the lengths seen in the dataset, and value as prior computed only from questions with corresponding length, for example:
$$3: [0.5, 0.3, 0.2],$$
$$4: [0.4, 0.3, 0.1, 0.2],$$
$$N: [0.2, 0.1, 0.1, 0.04, 0.04, 0.01, ...]$$

To supplement the lengths with lower datapoint, we take prefix of the longer priors, then *normalize* to unit vector, and use as auxiliary datapoints to help computing for shorter priors, for example a 10-option prior (prior computed from 10-option question) can be used to help computing priors for 3-option question:
$$[0.12, 0.2, 0.05, 0.17, 0.04, 0.01, 0.01, 0.02, 0.3, 0.2]$$

$$\downarrow$$

$$[0.32, 0.54, 0.14]$$

We take the first 3 numbers corresponding to "ABC" of a 3-option question, then normalize it to the unit vector with the same probability distribution as the other 3-option priors. Similarly, this 10-option prior can also be used to compute priors for any shorter lengths.

Lastly, because Choice Funnel will remove the selected option from the option set, the option IDs (ABCD) would not be continuous. Because the prior vector can only work with a continuous option set, we must **rebalance the option IDs**. For example, "ACDE" ("B" is removed) will be rebalanced to "ABCD".

### K.3 Conclusion and Takeaways

Once we have done this process we should have a large enough population to compute accurate priors for most lengths. One limitation is that this adaptation does not help much if we don't have enough questions for longer lengths in our dataset, though this is not the case for SATA-Bench, which contains 21.88% data for its longest 15-option question. One potential solution is to use synthetic datasets to backfill longer-option questions, since the original work showed that the prior is transferable. We leave this for future work.

## L   Experiment Setup for Choice Funnel

We chose a fixed *90%* confidence threshold as the stopping condition (ii) in Choice Funnel for **all models**. While this initial parameter selection was chosen for its simplicity, later evaluation indicated that it yielded sufficiently robust performance. Consequently, we did not pursue further investigation into more granular threshold adjustments. It also demonstrates that the algorithm is generalizable to other models without careful calibration.

The first baseline method *first token* sets a fixed threshold so that any option with a probability above the threshold is selected, and this should be the lower bound of the performance. *First token debiasing* can be used to find out if the popular strategy used to solve the MCQ questions is transferable to the SATA questions in terms of minimizing the impact of the selection bias. Lastly, we expect *yes/no* to be a competitive baseline given that it processes each choice separately.

**Prompts.** To reduce the bias introduced by prompt design and emphasize the impact of the method itself, we choose prompts for all methods with minimal engineering effort and mainly capture the essential components: *paragraph, question and choices*. The complete prompts are given in Appendix H.

**Models.** Our study focuses on the causal, decoder-only LLMs since this architecture has become the dominant choice for modern LLMs. We experiment with *7 LLMs from Table 2 under Probability Based Retrieving* which are all popular open-source models on the HuggingFace website, and we can access their output probabilities: DeepSeek R1 Distilled LLAMA 8B [14], Qwen2.5 14B [54], Ministral 8B [50], Phi 3 7B [1], Phi 4 mini reasoning [2], Bloomz 7B [35], and Llama 3.1 8B [52].

## M   Ablation Study for Choice Funnel

### M.1   "I don't know" performs worse than "None of the above"

Table 11: Performance comparison of Choice Funnel using "None of the Above" versus "I don't know" options.

| Method | EM↑ | Precision↑ | Recall↑ | JI↑ | SPD↓ | CtDifAbs↓ | CtAcc↑ | InfCost↓ |
|---|---|---|---|---|---|---|---|---|
| Phi3-7B + *nota* | **0.292727** | **83.27** | 70.24 | 61.85 | 3.47 | **1.42** | **0.38** | **6339** |
| Phi3-7B + *idk* | 0.281818 | 80.92 | **73.25** | **62.22** | **2.35** | 1.48 | 0.36 | 6667 |
| Llama3-8B + *nota* | **0.198788** | **78.69** | 56.19 | **50.36** | 7.74 | **1.66** | **0.33** | **4975** |
| Llama3-8B + *idk* | 0.176364 | 75.50 | **58.03** | 49.55 | 7.74 | 1.69 | 0.32 | 5066 |
| Bloomz-7B + *nota* | **0.201818** | **66.62** | 54.90 | **46.15** | 17.78 | **1.71** | **0.32** | **5440** |
| Bloomz-7B + *idk* | 0.180000 | 65.55 | **55.76** | 45.53 | **16.45** | 1.76 | 0.31 | 5528 |

We compared two commonly employed auxiliary response options in traditional survey science domain [45]: 'I don't know' (*IDK*) and 'None of the above' (*NOTA*), examining their effectiveness as *Choice Funnel* stopping condition. Based on an ablation study on Table 13, *NOTA* yields consistently

better performance. When using *IDK*, we observe **noticeable increase in *InfCost* and result in worse Count Bias** (*CtDifAbs* and *CtAcc*), which means **model tends to over select number of options**, indicating that the model would rather select a wrong answer than saying "I don't know". This is potentially related to RLHF process, where the model is trained to generate answers that are more favorable to humans.

## M.2 Ablation on Choice Funnel Components

Table 12: Ablation study demonstrating that PriDe token debiasing effectively mitigates unselection bias.

| Method | EM↑ | Precision↑ | Recall↑ | JI↑ | SPD↓ | CtDifAbs↓ | CtAcc↑ | InfCost↓ |
|---|---|---|---|---|---|---|---|---|
| Phi3-7B + *debiasing only* | 1.76 | 67.92 | 28.24 | 27.47 | 175.24 | 2.50 | 0.05 | **2534** |
| Phi3-7B + *CF only* | 26.00 | 80.84 | 70.08 | 60.33 | 4.17 | 1.44 | 0.35 | 6436 |
| Phi3-7B + *CF + debiasing* | **29.27** | **83.27** | **70.24** | **61.85** | **3.47** | **1.42** | **0.38** | 6339 |
| Llama3-8B + *debiasing only* | 7.58 | 62.83 | 32.28 | 30.38 | 151.74 | 2.34 | 0.14 | **2534** |
| Llama3-8B + *CF only* | 17.45 | 76.37 | 50.84 | 46.74 | 10.12 | 1.67 | **0.34** | 4380 |
| Llama3-8B + *CF + debiasing* | **19.88** | **78.69** | **56.19** | **50.36** | **7.74** | **1.66** | 0.33 | 4975 |
| Bloomz-7B + *debiasing only* | 7.09 | 59.07 | 38.41 | 32.05 | 149.17 | 2.19 | 0.15 | **2534** |
| Bloomz-7B + *CF only* | 16.36 | 66.10 | 48.26 | 42.66 | 23.09 | **1.65** | **0.35** | 4469 |
| Bloomz-7B + *CF + debiasing* | **20.18** | **66.62** | **54.90** | **46.15** | 17.78 | 1.71 | 0.32 | 5440 |

We conducted an ablation study on the two sub-components of Choice Funnel: token debiasing (*"debiasing only"*) and iterative selection (the process of iteratively selecting options until a stopping condition is met, denoted as *"CF only"*). The analysis is performed on 3 open-source models. When comparing *"CF only"* to the complete *"CF + debiasing"*, the observed increase in SPD metric demonstrates that **token debiasing effectively mitigates unselection bias**, yielding better performance. Nevertheless, the comparison between *"debiasing only"* and *"CF only"* reveals that **our novel iterative selection component contributes more substantially to overall performance improvements.**

## M.3 Ablation on Choice Funnel Stopping Condition

Table 13: Ablation study on the two stopping conditions in Choice Funnel, showing that combining both yields the best performance.

| Method | EM↑ | Precision↑ | Recall↑ | JI↑ | SPD↓ | CtDifAbs↓ | CtAcc↑ | InfCost↓ |
|---|---|---|---|---|---|---|---|---|
| Phi3-7B + *thresholding only* | 3.82 | 65.00 | 74.84 | 48.93 | 3.37 | 2.22 | 0.13 | 7416 |
| Phi3-7B + *NOTA only* | 29.21 | 77.07 | **85.63** | **68.00** | **0.69** | **1.20** | 0.37 | 9380 |
| Phi3-7B + *thresholding + NOTA* | **29.27** | **83.27** | 70.24 | 61.85 | 3.47 | 1.42 | **0.38** | 6339 |
| Llama3-8B + *thresholding only* | 0.89 | 71.92 | 52.22 | 44.12 | 10.53 | 1.74 | 0.27 | 4564 |
| Llama3-8B + *NOTA only* | 19.51 | 69.22 | **85.77** | **60.09** | 2.24 | 1.94 | 0.25 | 10212 |
| Llama3-8B + *thresholding + NOTA* | **19.88** | **78.69** | 56.19 | 50.36 | 7.74 | **1.66** | **0.33** | 4975 |
| Bloomz-7B + *thresholding only* | 9.94 | 64.47 | 48.93 | 40.77 | 22.50 | 1.72 | 0.29 | 4506 |
| Bloomz-7B + *NOTA only* | 12.24 | 55.60 | **89.57** | **52.81** | 12.82 | 3.31 | 0.17 | 13758 |
| Bloomz-7B + *thresholding + NOTA* | **20.18** | **66.62** | 54.90 | 46.15 | 17.78 | **1.71** | **0.32** | 5440 |

We conducted an ablation study to evaluate the relative importance of our two proposed stopping conditions in Choice Funnel. The results demonstrate that Choice Funnel achieves optimal performance when both conditions are applied in combination. Notably, the "None of the above" (*NOTA*) condition emerged as the more influential factor, suggesting that models can reliably identify when no correct answers remain among the provided options.

# N    Positional Bias Under Randomized Answer Orderings

**Does the benchmark include randomized answer orderings?**    No. In the main benchmark, each question's answer choices appear in a fixed, canonical order. To quantify the extent to which large language models (LLMs) rely on this implicit positional cue, we ran an auxiliary study in which the answer choices for every question were *randomly permuted* (e.g. A B C → C A B). We then compared model performance on the permuted dataset to its performance on the original version.

**Setup.**    Three representative models were evaluated Nova Pro, Claude 3 Haiku, Llama 3.1 405B.

All hyper-parameters, prompts, and decoding settings were kept *identical* to the main benchmark; only the answer order was shuffled once per question. Table 14 reports the *difference* (*permute–original*) for each metric, so negative values indicate a drop in performance and positive values indicate an increase. [†] **CtDif** is shown with a downward arrow even though its baseline values are negative; a more negative CtDif therefore indicates a larger absolute mismatch in option counts.

Table 14: Change in evaluation metrics after randomly reordering answer choices. Performance metrics are expected to **increase** (↑) while bias metrics are expected to **decrease** (↓).

| Model | EM ↑ | Precision ↑ | Recall ↑ | JI ↑ | RStd ↓ | RSD ↓ | SPD ↓ | CtDif[†] ↓ | CtDifAbs ↓ |
|---|---|---|---|---|---|---|---|---|---|
| Nova Pro | −2.26 | −0.05 | −4.13 | −3.46 | +8.98 | −0.03 | +0.20 | −0.13 | −0.66 |
| Claude 3 Haiku | −24.06 | −34.69 | −34.28 | −35.31 | +6.06 | +0.17 | +0.12 | −0.07 | −0.51 |
| Llama 3.1 405B | −3.80 | −3.90 | −4.71 | −5.22 | +9.73 | −0.20 | +0.25 | −0.18 | −0.71 |

**Findings.** All three models suffer performance degradation when answer choices are shuffled, with **Claude 3 Haiku** exhibiting the sharpest decline (–24 EM, –35 JI). Selection/count-bias metrics (RStd, SPD, CtDifAbs) *increase* for every model except RSD on Nova Pro, confirming heightened positional bias.

**Discussion.** These results suggest that current LLMs implicitly learn positional heuristics from training data in which answer orders are fixed. Breaking this assumption makes the models less certain and more prone to biased guessing. Future work should examine (i) whether fine-tuning on randomly ordered choices mitigates the effect, and (ii) how pronounced the bias is for other model families and task domains.

# O  Per-Dataset Performance Breakdown

We report detailed bias metrics for different task categories in Table 15. The News dataset has the lowest selection bias, while Reading Comprehension exhibits the highest. For count bias, Toxicity shows the smallest difference, and Biomedicine has the largest. Notably, News has significantly lower selection and count biases compared to other datasets (p-values: 0.03 for SPD and $3.8 \times 10^{-5}$ CtDifAbs, T-test). All datasets show negative count difference, confirming underprediction and the presence of count bias in SATA questions.

Table 15: Breakdown of Bias metrics by subject. Lower values are better for all metrics.

| Task | RStd ↓ | RSD ↓ | SPD ↓ | CtDif | CtDifAbs ↓ |
|---|---|---|---|---|---|
| Reading Comprehension | 19.29 ± 7.59 | 0.20 ± 0.10 | 1.53 ± 1.39 | -0.68 ± 0.42 | 0.85 ± 0.35 |
| Toxicity | 7.13 ± 2.83 | 0.11 ± 0.07 | 0.48 ± 0.56 | -0.05 ± 0.44 | 1.28 ± 0.16 |
| News | 4.32 ± 3.16 | 0.08 ± 0.19 | 0.12 ± 0.23 | -0.09 ± 0.25 | 0.32 ± 0.19 |
| Biomedicine | 6.66 ± 2.37 | 0.15 ± 0.14 | 2.90 ± 3.60 | -1.71 ± 0.96 | 2.22 ± 0.67 |
| Laws | 5.75 ± 4.17 | 0.13 ± 0.16 | 1.54 ± 3.43 | -1.00 ± 0.87 | 1.36 ± 0.75 |
| Events | 7.15 ± 4.14 | 0.13 ± 0.19 | 0.85 ± 1.02 | -0.28 ± 0.77 | 1.08 ± 0.30 |

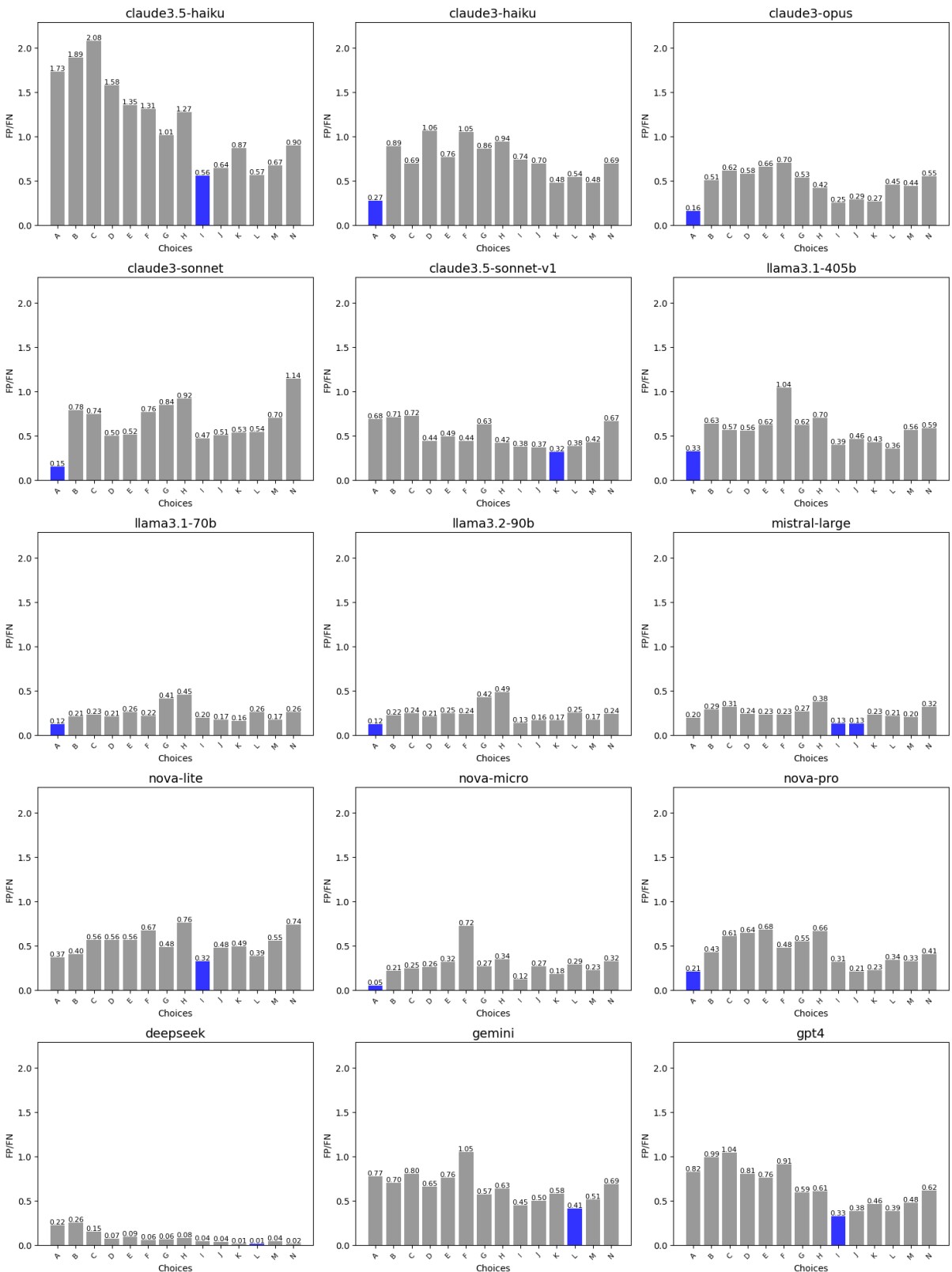

Figure 9: Ratio of false positive rate to false negative rate per label for each evaluated LLM.

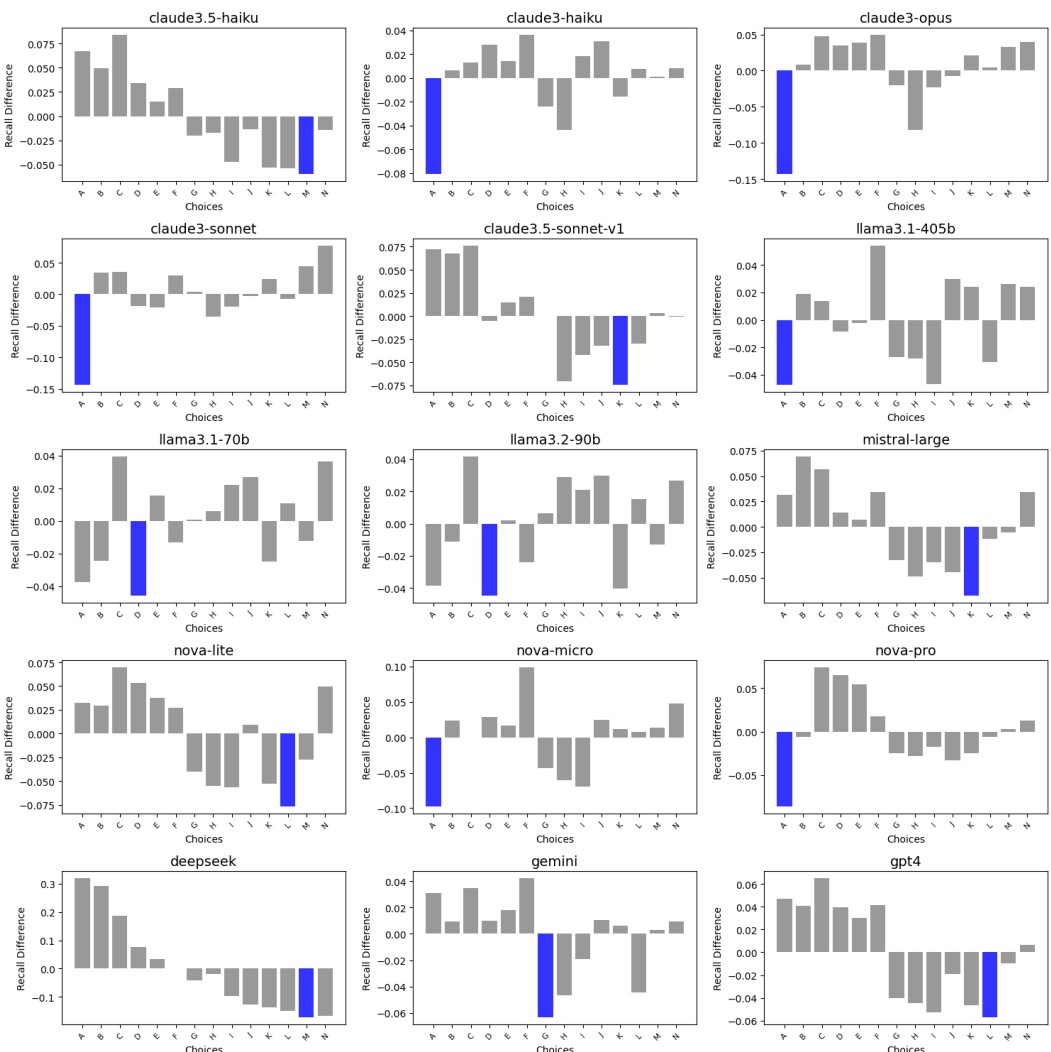

Figure 10: Recall score per label (Y-axis), normalized by subtracting the model's average recall. Most models exhibit at least one label with significantly lower recall than the rest.

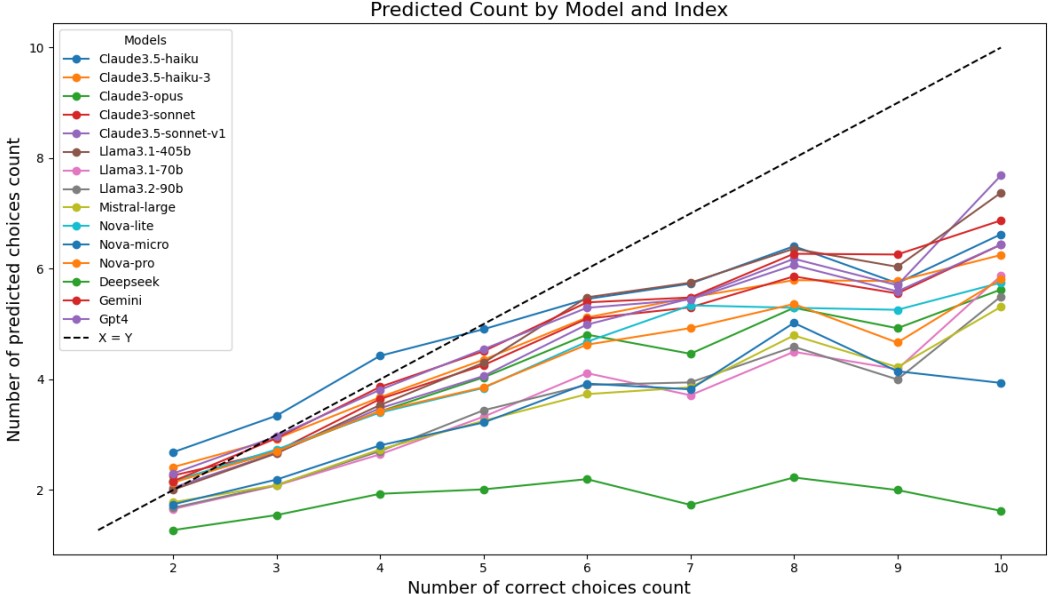

Figure 11: Relationship between predicted and actual correct choice counts across models. Models generally under-select the correct number of answer choices. Y-axis represents the average number of choices selected by the model. X-axis represents the actual number of correct choices. A perfect model would align along the diagonal where X equals Y.

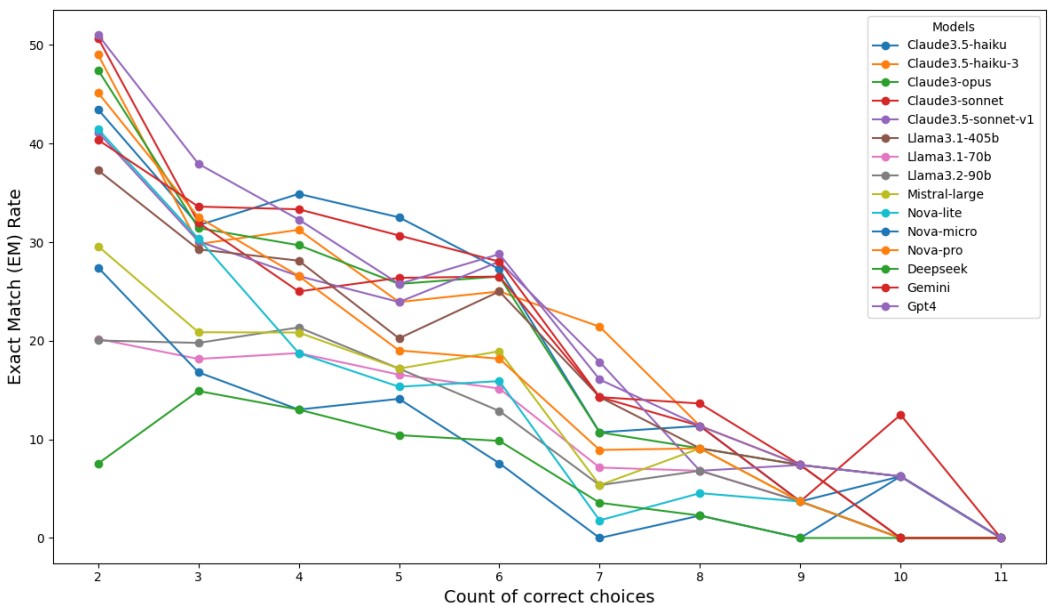

Figure 12: Relationship between Exact Match Rate and the number of correct choices. As the number of correct choices increases, the exact match rate decreases. None of the models achieve an exact match rate above 20% when the number of correct choices exceeds 7.

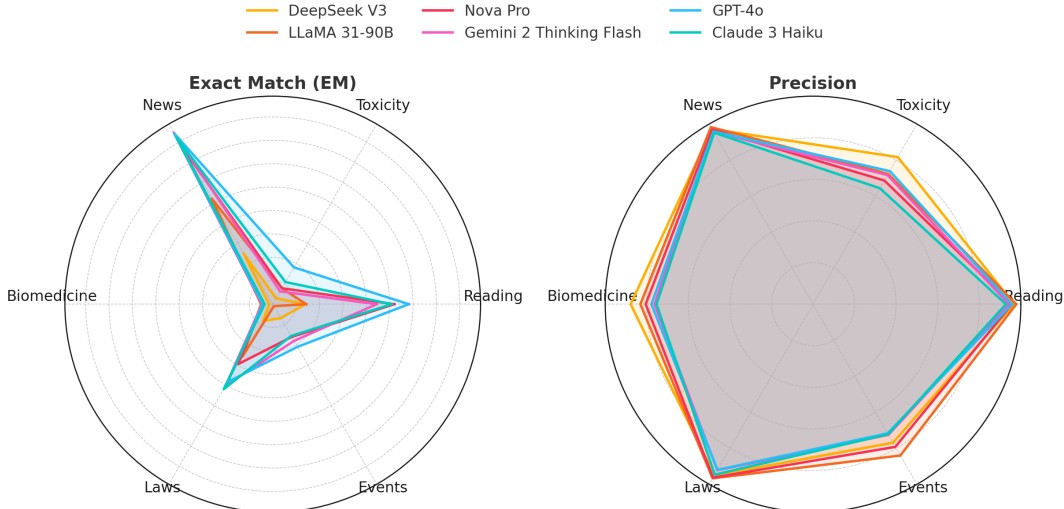

Figure 13: Performance breakdown of evaluated models across different source datasets.