# OpenReview forum: "SATA-BENCH: Select All That Apply Benchmark for Multiple Choice Questions"
_NeurIPS.cc/2025/Datasets_and_Benchmarks_Track — Submitted to NeurIPS 2025 Datasets and Benchmarks Track_

### Official Review · Reviewer_TePK · 2025-06-26

**Rating:** 4
**Confidence:** 3

**Summary:**

This paper introduces SATA-BENCH, the first dedicated benchmark for evaluating large language models (LLMs) on "Select All That Apply" (SATA) multiple-choice questions across diverse domains including reading comprehension, law, and biomedicine. The authors argue that existing benchmarks, primarily single-answer multiple-choice tasks, overlook this crucial real-world capability. Their evaluation of 30 models reveals a significant performance gap, with the strongest models achieving only 41.8% exact match, indicating LLMs' inability to reliably identify all correct answers. The paper attributes this weakness to two core challenges: selection bias (favoring certain choices) and count bias (failing to predict the correct number of answers). To address these, they propose Choice Funnel, a decoding strategy that combines token debiasing with adaptive thresholding, demonstrating improved exact match performance and reduced inference cost.

**Dataset Code Accessibility:**

Yes

**Dataset Code Comments:**

The dataset has been released.

**Ethical Considerations:**

No, there are no or only very minor ethics concerns

**Final Justification:**

My concerns are addressed so I maintain the original positive rating.

**Limitations Weaknesses:**

- **Potential for Memorization**: The authors acknowledge that they cannot fully rule out the possibility of LLMs having been exposed to parts of the source datasets during pretraining, especially for proprietary models. While this is a common challenge, it means the benchmark might not entirely isolate true reasoning from memorization for all models.

- **Limited Domain Coverage (Relative to Broader Benchmarks)**: While covering six diverse domains is good, the paper itself notes that the total number of domains is limited compared to larger-scale benchmarks like MMLU. Further expansion would be beneficial for broader generalization.

- **Acknowledged Label Correctness Issues**: Despite rigorous human validation, the authors acknowledge that perfect correctness of all labels cannot be guaranteed due to the inherent complexity and potential ambiguities in domains like biomedicine and law. This suggests a minor margin for error in the ground truth.

**Strengths Contributions:**

- **Novel Benchmark Type**: SATA-BENCH is a pioneering effort in creating a dedicated benchmark for "Select All That Apply" questions, a critical yet underexplored capability for LLMs in real-world scenarios. This fills a notable gap in current LLM evaluation paradigms, which often focus on single-answer multiple-choice formats.

- **Diverse Domain Coverage**: The benchmark spans six diverse and challenging domains including reading comprehension, biomedicine, and law, which contributes to its robustness and relevance across various knowledge areas.

- **In-depth Analysis of LLM Biases**: The paper effectively diagnoses two fundamental limitations of LLMs: selection bias and count bias. This detailed analysis provides valuable insights into why LLMs struggle with SATA questions, moving beyond mere performance reporting to explain underlying model behaviors.

- **Effective Decoding Strategy (Choice Funnel)**: The proposed Choice Funnel decoding strategy directly addresses the identified biases, demonstrating a notable improvement in exact match performance and efficiency. This provides a practical solution and a framework for improving multi-answer reasoning.

- **Extensive Model Evaluation**: The comprehensive evaluation of 30 open-source and proprietary models provides a thorough understanding of the current state-of-the-art in multi-answer reasoning, establishing strong baselines for future research.

---

> ### Author Rebuttal · Authors · 2025-07-30
>
> Thank you for your constructive review. We sincerely appreciate your recognition of the novelty and diversity of SATA-BENCH, as well as your appreciation of our in-depth analysis of selection and count bias, the Choice Funnel decoding strategy, and our comprehensive model evaluation. We address your concerns below:
>
> **W1 Potential for Memorization: the benchmark might not entirely isolate true reasoning from memorization for all models.**
> - While we acknowledge that it is difficult to rule out memorization entirely, we applied the open-source contamination detection pipeline from Li et al. [1] to evaluate potential data leakage. Specifically, we used the Bing Search API to probe for verbatim matches of each question in SATA-BENCH. For each question, 20 query variants were tested to maximize detection capability. Any pages retrieved were then cross-checked against the Common Crawl index to determine whether they were likely seen during pretraining.
>
> - None of the questions in our benchmark were flagged as contaminated, providing strong evidence that they were not indexed by Common Crawl or directly retrievable via public search engines. This makes our dataset unlikely to overlap with the pretraining corpora of major LLMs, thus reducing concerns about memorization.
>
> **W2 Limited Domain Coverage (Relative to Broader Benchmarks)**
> - We designed our benchmark to reflect realistic applications that LLMs are likely to encounter. The six current domains were chosen based on practical relevance:
>     - LLM-assisted annotation: Tasks like content moderation, compliance checks, or topic classification often require assigning multiple labels. This motivates our inclusion of datasets like Toxicity, News, and Events.
>     - Education and self-study: In domains like law or medicine, users may ask LLMs which principles, diagnoses, or statutes apply to a scenario. (Law and Biomedicine datasets)
>     - Document summarization or retrieval: Extracting all relevant facts from text is a multi-answer task. (Reading Comprehension dataset)
> - These domains together provide broad cognitive and linguistic coverage. Moreover, SATA-BENCH complements existing benchmarks (which focus on single-answer formats or math/coding problems) by specifically targeting the **multi-answer reasoning challenge**.
>
> **W3: Acknowledged Label Correctness Issues: Despite rigorous human validation, the authors acknowledge that perfect correctness of all labels cannot be guaranteed due to the inherent complexity and potential ambiguities in domains like biomedicine and law. This suggests a minor margin for error in the ground truth.**
> - All annotators were professional annotators with prior experience working on multi-label tasks and domain-specific content (including 6 domains that is covered by our benchmark such as medicine and law).
> - The Human Standard Operation Procedure (SOP) is drafted by technical writer who is professional in writing annotation instructions.
> - During annotation, annotators were provided with LLM-generated answers alongside the original label set, and were encouraged to consult trusted sources (e.g., via web search) when needed, as simplified instruction is documented in Appendix B.3.
> - Since each question may have multiple correct answers, we report pairwise agreement between the first two annotators, which was 91.22%. After filtering out low-quality and ambiguous questions by the third annotators, the agreement rate increased to 96.51% in our reported dataset.
> - Since a third annotator reviewed all cases where the first two annotators disagreed, the actual error rate is expected to be significantly lower than 3.49%.
> - The complete process is described in detail in Appendix B.3. While a small margin for error is inevitable in such domains, we believe our multi-stage annotation and validation pipeline provides high-quality ground truth suitable for rigorous LLM evaluation.
>
> [1] Li, Yucheng, Frank Guerin, and Chenghua Lin. "An open source data contamination report for large language models." arXiv preprint arXiv:2310.17589 (2023).

---

> > ### Comment · Reviewer_TePK · 2025-08-04
> >
> > Thanks for your response. My concerns have been addressed. I will maintain my positive rating.

---

> > > ### Author Response · Authors · 2025-08-05
> > >
> > > Thank you once again for your thorough review and insightful feedback! We are pleased that **our additional experiments and explanations have addressed all of your stated concerns.**
> > >
> > > We are curious if, despite addressing your concerns, there are any remaining aspects of the work that you feel could be further strengthened, or if there are any additional considerations that would lead to a higher evaluation of our contribution. Thank you for your time and consideration.

---

### Official Review · Reviewer_j7Yo · 2025-07-03

**Rating:** 5
**Confidence:** 3

**Summary:**

This paper introduces SATA, a multi-answer benchmark dataset designed to evaluate LLMs’ reasoning capabilities. The authors conduct a comprehensive assessment of 30 LLMs on SATA and uncover two key challenges in multi-answer inference: selection bias and count bias. To address these issues, they propose the Choice Funnel algorithm, which effectively improves multi-answer reasoning performance by mitigating both biases.

**Additional Feedback:**

1. Figure 3: Most metrics follow a Gaussian-like distribution, except for the Flesch Reading Ease Score. The authors are encouraged to explain why this distribution deviates from the others ?
2. It unclear whether annotator agreement was evaluated during the annotation process. Providing inter-annotator agreement statistics (e.g., Cohen’s kappa or Fleiss’ kappa) would strengthen the reliability of the dataset.
3. Figures 5 and 6 appear in the appendix but are referenced without clear indication in the main text. To avoid confusion, the authors should explicitly note that these figures are located in the appendix.
4. The manuscript should clarify how malformed or incorrect JSONL outputs are handled during evaluation. This is particularly important for ensuring robustness in automated reasoning pipelines and reproducible benchmarking.

**Dataset Code Accessibility:**

Yes

**Ethical Considerations:**

No, there are no or only very minor ethics concerns

**Final Justification:**

The authors response has addressed my concerns, and I give them a positive rating.

**Limitations Weaknesses:**

1. All data in SATA are collected from existing datasets, which may limit the contribution.
2. Selection bias and count bias are well-documented issues in the LLM research community.

**Strengths Contributions:**

1. The authors conduct a comprehensive evaluation of 30 LLMs on their proposed SATA benchmark and empirically verify key challenges in the multi-answering tasks:  selection bias and count bias in current LLMs.
2. To address these challenges, the authors propose an algorithm named Choice Funnel, which improves answer selection by reducing both selection and count bias in multi-answering task.

---

> ### Author Rebuttal · Authors · 2025-07-30
>
> Thank you for your constructive review. We sincerely appreciate your recognition of our comprehensive evaluation and the proposed **Choice Funnel algorithm**. Below we address your specific comments and suggestions.
>
> **W1: All data in SATA are collected from existing datasets, which may limit the contribution.**
> - While the source datasets do exist in the context of traditional multi-label classification, they are not directly usable for LLM evaluation due to several key limitations—such as bag-of-words formats, label sparsity, and lack of linguistic structure.[1]
>
> - We applied a *multi-stage transformation pipeline* involving *question reformulation, distractor synthesis, human validation of clarity, and human-verified correctness of the answer sets*. These steps increase linguistic realism, reasoning complexity, and label quality, making the resulting benchmark challenging, diverse, and well-suited for evaluating multi-answer reasoning in LLMs.
>
> **W2: Selection bias and count bias are well-documented issues in the LLM research community.**
> - We agree that selection bias has been studied in the context of single-answer MCQs. However, we are the first to demonstrate that selection bias manifests in SATA-style questions despite the increased answer space and format. We further discover unselection bias (a form of selection bias where models consistently avoid certain options) in SATA-style questions.
>
> - Also, we introduce a new bias type (called Count Bias) unique to SATA, where models consistently fail to predict the correct number of answers.
>
> - To our knowledge, no prior work has studied unselection bias and count bias, and our benchmark provides both new diagnostic tools and metrics (e.g., SPD, CountAcc) to study them.
>
> **Q1: Most metrics follow a Gaussian-like distribution, except for the Flesch Reading Ease Score. The authors are encouraged to explain why this distribution deviates from the others ?**
>
> - **Right-Skew Toward Challenging Texts:** As noted in Section 2.3(iv), we intentionally curated SATA-BENCH with questions that simulate real-world professional tasks. Consequently, 76% of questions fall within the 20–60 range (interpreted as “standard to very hard” in readability), producing a right-skewed shape.
>
> - **FRE's Non-Symmetric Nature:** Unlike Gunning Fog or semantic similarity scores, FRE scores are bounded (typically between 0 and 100). It is calculated using the formula:
> 206.835 - (1.015 x Average Sentence Length) - (84.6 x Average Syllables per Word)
> This formula disproportionately penalizes small increases in complexity, especially in domains like law, biomedicine, or news, causing real-world texts to naturally cluster at lower readability scores. The result is a right-skewed distribution with a heavy concentration around 20–60, which is clearly observed in our benchmark (Figure 3, bottom left).
>
> - **Alignment with Benchmark Goals:** This skew is consistent with our design objective to span a broad range of reading difficulty levels, from high-school to graduate-level comprehension, while avoiding trivial or ambiguous content.
>
> **Q2: It unclear whether annotator agreement was evaluated during the annotation process. Providing inter-annotator agreement statistics (e.g., Cohen’s kappa or Fleiss’ kappa) would strengthen the reliability of the dataset.**
> - Due to the multi-answer and multi-choice format, we can only report pairwise agreement rate. The first two annotators' pre-filtering agreement rate was 91.22%. After removing ambiguous or low-confidence questions by the third annotator, the final dataset reached a 96.51% agreement rate. These statistics confirm that our annotation process is rigorous and yields high inter-annotator reliability.
> - We agree with the reviewer and will share those statistics in the revised draft.
>
> **Q3: Figures 5 and 6 appear in the appendix but are referenced without clear indication in the main text. To avoid confusion, the authors should explicitly note that these figures are located in the appendix.**
> - Thank you for point this out. We agree this could cause confusion. We will explicitly reference Figures 5 and 6 as belonging to the Appendix in the revised main text and ensure all cross-references are clearly marked.
>
> **Q4: The manuscript should clarify how malformed or incorrect JSONL outputs are handled during evaluation. This is particularly important for ensuring robustness in automated reasoning pipelines and reproducible benchmarking.**
>
> - We agree that robustness to malformed outputs is essential for reproducible LLM evaluation. Our approach is as follows: For cases where no match is found in the JSON-formatted output, we employ Claude 3 Haiku to extract the final answer table. We then verify these outputs using annotators in Amazon GroundTruth, ensuring that the final labels are correct.
> - This process is outlined in Appendix I, and we will make this explicit in the revised main text to clarify our robust error-handling and evaluation pipeline.
>
> Thank you for all those insightful and thoughtful questions, which is extremely helpful for us to improve our manuscript!
>
> [1] Multi-Label Classification Dataset Repository. https://www.uco.es/kdis/mllresources/

---

### Official Review · Reviewer_YYAZ · 2025-07-03

**Rating:** 4
**Confidence:** 4

**Summary:**

This paper introduces SATA-BENCH, a multi-domain, human-validated benchmark dataset for systematically evaluating large language models on multi-answer selection tasks (Select All That Apply, SATA). The authors construct a total of 1,604 high-quality SATA questions covering six real-world application domains.

Based on this benchmark, the authors conduct a systematic evaluation of 30 mainstream LLMs, including both open-source and proprietary models. The results show that even the strongest model achieves only 41.8% accuracy under the strict exact match metric, revealing the widespread presence of selection bias and count bias in LLMs when handling multi-answer selection tasks.

To address these issues, the authors propose the Choice Funnel algorithm, which combines token debiasing with adaptive threshold control and iteratively selects answers to effectively improve accuracy in multi-answer selection tasks. Experiments demonstrate that Choice Funnel improves exact match performance by approximately 29% on seven mainstream open-source models.

Overall, this work reveals the limitations of LLMs in realistic multi-answer reasoning, proposes an effective improvement strategy, and provides a standardized, publicly available evaluation benchmark for future research.

**Dataset Code Accessibility:**

Yes

**Ethical Considerations:**

No, there are no or only very minor ethics concerns

**Final Justification:**

The authors have addressed most of my concerns in the rebuttal with additional results and comparison. I have raised my score to lean towards acceptance.

**Limitations Weaknesses:**

- The work lacks comparison with existing similar datasets, making the claimed novelty unconvincing. There are already multi-answer datasets; this is not the first of its kind. Compared to other similar datasets, the proposed dataset is small in scale. Moreover, the authors themselves acknowledge that “we cannot guarantee perfect correctness of all labels despite triple human annotation and agreement filtering.”, and no error rate is reported. Given the small size and potential inaccuracies, the contribution remains unclear.

- The label design is inconsistent and contains significant redundancy, with no label normalization applied. For example, the answer groups include numerous semantically similar or overlapping labels such as [“Against women activities”, “Against”], [“U.S. Government”, “Us”, “The U.S. Government”, “American citizens”], [“Before 1945”, “1940”]. Given that the experiments emphasize LLM selection bias and count bias, it is unclear whether part of the observed bias is amplified by the dataset's label design rather than being inherent to the models.

- The reliability of the Count Bias measurement is questionable. As mentioned above, the dataset itself contains redundant or semantically similar answers within the correct answer set. Thus, when a model selects fewer answers, it may be due to incorrect predictions of semantically similar but wrong options, rather than a genuine omission of correct answers.

- The impact of option length on model performance is not discussed. Was there any control over the length of answer options? LLMs may exhibit a significant preference for shorter options.

- The task selection process for SATA-BENCH is unclear. For example, MMLU contains 57 subjects, yet only law and medical domains were chosen. The rationale behind this selection lacks justification. The qualifications and backgrounds of the annotators are not provided. For domain-specific datasets, such as those in law and medicine, expert annotation is generally expected.

- The proposed Choice Funnel method lacks thorough analysis. It is unclear why it sometimes fails to reduce or even amplifies bias.
It is mentioned that the order of answer options may influence model performance. However, in the construction of SATA-BENCH and in the experiments, no control or discussion of this factor is provided.

- The dataset construction process involves considerable randomness, especially in distractor sampling and option order shuffling. No analysis is provided on the stability of results under different random configurations. Given that the final dataset only contains 1604 instances, such randomness could substantially affect the overall distribution and experimental results.

- There are spelling errors in the paper, such as "Geimini 2.5 Think" in the table on page 5.

**Strengths Contributions:**

- The authors introduce SATA-BENCH, a curated benchmark specifically designed to evaluate LLM performance on realistic multi-answer "Select All That Apply" (SATA) tasks, addressing a notable gap in existing evaluation resources.

- The dataset covers six diverse domains—including law, biomedicine, and reading comprehension—with 1,604 human-validated questions, ensuring both task variety and rigorous quality control.

- The study presents comprehensive evaluation to date of 30 proprietary and open-source LLMs on SATA tasks, revealing significant model limitations such as selection bias and count bias, with the best exact match score remaining 41.8%.

- The proposed Choice Funnel decoding algorithm, combining token debiasing and adaptive thresholding, effectively mitigates these biases.

- The paper is clearly written, and provides sufficient experimental details, with tables and figures that comprehensively illustrate dataset characteristics, model comparisons, and ablation results.

---

> ### Author Rebuttal · Authors · 2025-07-30
>
> Thank you for your thoughtful review and going through the dataset! We appreciate your recognition of the diversity, variety, and quality control of our dataset. Your recognition of the effectiveness of Choice Funnel decoding algorithm is encouraging.
>
>
> **W1: lacks comparison with existing similar datasets. There are already multi-answer datasets; this is not the first of its kind.**
>
> - Although other multi-answer datasets exist, they typically serve multi-label classification and are not designed for LLM-based multi-answer reasoning. Most prior works use traditional deep learning methods to solve multi-label classification [1]. Thus, they use bag-of-words formats and are not structured as text-based MCQs, which makes them impossible for LLM evaluation[2].
>
> - Prior work has not evaluated LLMs' performance on SATA questions in multiple domains.[3] We have broadened the scope of the LLM evaluation to make it work with various domains. The novelty of our benchmark is also recognized by other two reviewers.
>
> | Benchmarks | Domain Coverage | Data Format | Test on LLMs  |
> |:---:|:---:|:---:| :---:|
> | MultiRC[3] | Single | Text | Yes |
> | Multi-Label Classification Dataset[2]  | Multiple | Bag of words | No |
> | SATA Bench  | Multiple | Text | Yes |
>
> **W2: dataset is small in scale.**
> - While SATA-BENCH contains 1,604 high-quality questions, we have also released: (i) An extended 7,983-question dataset, and (ii) A 1,570-question single-answer variant. (mentioned in NeurlPS Paper Checklist Q5, 11157 questions in total)
>
> **W3:  no human labeling error rate is reported. Its accuracy is not clear. The qualifications and backgrounds of the annotators are not provided.**
> - *All annotators were professional annotators with prior experience working on multi-label tasks and domain-specific content (including 6 domains that are covered by our benchmark, such as medicine and law).*
> - The Human Standard Operation Procedure (SOP) is drafted by a technical writer, whose main job is writing annotation instructions. The simplified instruction is documented in Appendix B.3.
> - Since each question may have multiple correct answers, we report pairwise agreement between the first two annotators, which was 91.22%. After filtering out low-quality and ambiguous questions by the third annotators, *the agreement rate is 96.51% in our reported dataset.*
> - Since a third annotator reviewed all cases where the first two annotators disagreed, the actual error rate is expected to be significantly lower than 3.49%.
>
> **W4: The label design is inconsistent and contains significant redundancy. The reliability of the Count Bias measurement is questionable.**
> - To ensure the diversity of the dataset labels, we ensure that our answer group has labels with different similarity. To assess label redundancy, we encoded labels using SentenceTransformer (all-MiniLM-L6-v2) and computed pairwise similarities. The mean maximum similarity across label sets is 0.473, with standard deviation 0.206. This confirms a mix of semantically similar and distinct labels. The top 10 percentile score is 0.786 and the bottom 10 percentile score is 0.235. This shows that our dataset has diverse labels with similar percentage of semantically similar and dissimilar labels.
> - Count bias increased after removing similar-label questions, suggesting that LLMs sometimes use semantic similarity to infer related correct answers. We remove all questions that have label pairs with similarity score over 0.786. We then recalculated count bias related metrics across all closed-source models. CtDif is lower and CtDifAbs get higher. This means that removing similar labels in question actually increase the number of count bias. We suspect that is due to the fact that LLM can reasoning through similar labels and use those labels' similarity to identify all correct answers.
> | compare | CtDif↓ | CtDifAbs↓ | CtAcc↑ |
> |:---:|:---:|:---:|:---:|
> | original | -0.47 | 1.07 | 37.71 |
> | w/o similar labels questions  | - 0.51 | 1.12 | 37.41 |
>
> **W5: Was there any control over the length of answer options? LLMs may exhibit a significant preference for shorter options.**
> - Yes, we have selected a distractor that is similar in length as the correct option. 62.34% of examples have equal median lengths across correct and distractor options. Option length median is 2 for both correct option and distractor. A t-test comparing these lengths yields a p-value > 0.05, indicating *no significant difference between distractor option length and all option length.*
> - We agree that option length has negative effect on performance. We have splited all correct answers by white space as its length(“U.S. Government” -> length 2). We then calculate per label length recall, precision and f1 score for all reported closed-source models. We then run linear regression where X is length and Y is recall, precision and f1 score to see if the slope is decreasing. As the length increase, all metrics have decreased with p value less than 0.1. This shows that LLMs exhibit a preference for shorter options. *Our regression analysis confirms that LLM performance degrades with longer answer phrases, with significant negative slopes for recall, precision, and F1 score.* This could be an interesting research direction!
>
> | Metric| Trend | P value| Slope |
> |:---:|:---:|:---:|:---:|
> | Recall | Decreasing | 0.0521 | -0.0031 |
> | Precision | Decreasing | 1.34E-9 | -0.0047 |
> | F1 Score | Decreasing | 0.003 | -0.0035 |
>
> **W6: The task selection process for SATA-BENCH is unclear. For example, MMLU contains 57 subjects, yet only law and medical domains were chosen. The rationale behind this selection lacks justification.**
>
> - The six domains were selected based on practical, real-world tasks where identifying multiple applicable answers is critical:
>     - LLM-assisted annotation: Tasks like content moderation, compliance checks, or topic classification often require assigning multiple labels. This motivates our inclusion of datasets like Toxicity, News, and Events.
>     - Education and self-study: In domains like law or medicine, users may ask LLMs which principles, diagnoses, or statutes apply to a scenario. (Law and Biomedicine datasets)
>     - Document summarization or retrieval: Extracting all relevant facts from text is a multi-answer task. (Reading Comprehension dataset)
>
> - As recognized by Reviewer TePK, these scenarios highlight the practical importance of multi-answer reasoning. SATA serves as a structured proxy for multi-label classification, open retrieval, and checklist-style assessment. We explored free-form QA but found it difficult to match LLM outputs to structured ground truth.
>
> **W7: The proposed Choice Funnel method lacks thorough analysis. It is unclear why it sometimes fails to reduce or even amplifies bias.**
> - We conducted detailed ablation studies on each component of the Choice Funnel algorithm, as presented in Appendix Section M. To mitigate token-level biases, we incorporated a token debiasing step. Comparing the base Choice Funnel (CF only) with the debiased variant (CF + debiasing) in Appendix M.2, we observe a significant improvement in SPD, the metric for selection bias.
>
> - The yes/no prompting baseline—which evaluates each option independently—represents a strong bias-minimizing approach by eliminating positional and token ID effects. Despite this, Choice Funnel outperforms yes/no in terms of Count Accuracy (CtAcc) across all 7 models, and Selection Probability Divergence (SPD) in 4 out of 7 models. Most notably, it achieves the highest Exact Match scores across all models, while requiring 64.58% fewer forward passes—a substantial efficiency gain.
>
> - This efficiency-accuracy tradeoff is particularly important for SATA-style questions, which often contain large option sets (SATA-BENCH supports up to 15 answer choices per question). Thus, Choice Funnel offers both performance and scalability, making it a practical decoding solution for multi-answer reasoning tasks. The proposal of the Choice Funnel Algorithm has been recognized by all other 3 reviewers.
>
> **W8: The dataset construction process involves considerable randomness, especially in distractor sampling and option order shuffling. No analysis is provided on the stability of results under different random configurations.**
> - While distractor options are sampled randomly, they are drawn from a constrained, domain-specific pool designed to ensure semantic relevance and diversity. This approach maintains consistent quality across tasks. Additionally, option order is uniformly shuffled for all questions to simulate real-world MCQ formats and mitigate positional bias, as detailed in Appendix B.
> - To assess the stability of distractor sampling, we conducted a seed variability analysis using a subsample of the dataset. The table below reports confusion scores across multiple random seeds. We find that the mean and distribution of confusion scores remain statistically consistent, indicating that our results are robust to sampling randomness during distractor selection.
> | Seed | Mean | Std | Min | Max | Median |
> |:---:|:---:|:---:|:---:|:---:|:---:|
> | 0 |  0.2502 | 0.0416 | 0.1306 | 0.4646 | 0.2460 |
> | 1 |  0.2508 | 0.0429 | 0.1238 | 0.4406 | 0.2450 |
> | 2 |  0.2513 | 0.0426 | 0.1417 | 0.4475 | 0.2458 |
> | 3 |  0.2512 | 0.0428 | 0.1275 | 0.4160 | 0.2450 |
> | 4 |  0.2508 | 0.0425 | 0.1240 | 0.4248 | 0.2457 |
> | 42 |  0.2480 | 0.0412 | 0.1029 | 0.4125 | 0.2435 |
> | 100 | 0.2504 | 0.0417 | 0.1342 | 0.4018 | 0.2446 |
>
> We will revise the manuscript to make the above points explicit and fix the typo.
>
> [1] Tarekegn, Adane Nega, Mohib Ullah, and Faouzi Alaya Cheikh. "Deep learning for multi-label learning: a comprehensive survey." arXiv preprint arXiv:2401.16549 (2024).
>
> [2] Multi-Label Classification Dataset Repository. https://www.uco.es/kdis/mllresources/
>
> [3] Looking Beyond the Surface: A Challenge Set for Reading Comprehension over Multiple Sentences

---

> ### Author Response · Authors · 2025-08-05
>
> We sincerely appreciate the time and effort you have already dedicated to reviewing our work. As the discussion phase is approaching its end,, we would be grateful if you could take a moment to consider our responses. Please let us know if there are any remaining concerns or points that would benefit from further clarification. We hope that our responses have addressed the issues raised, and we would welcome any additional feedback you might have.

---

> ### Comment · Reviewer_YYAZ · 2025-08-07
> **thank for the responese**
>
> Thanks to the authors for the responeses, which addressed most of my concerns. I have also read the comments and rebuttal to other reviewers. I will increase my score. Good luck.

---

> > ### Author Response · Authors · 2025-08-08
> > **Thank you**
> >
> > Thank you once again for your thorough review and insightful feedback! Thank you for providing interesting new research direction for us! We are pleased that **our additional experiments and explanations have addressed most of your stated concerns.**

---

### Official Review · Reviewer_Ymcx · 2025-07-05

**Rating:** 4
**Confidence:** 4

**Summary:**

The paper introduces a new benchmark specifically designed to evaluate LLM on "Select All That Apply" (SATA) multiple-choice questions. Unlike traditional multiple-choice evaluations that focus on single-answer selection, SATA-BENCH challenges LLMs to identify all correct answers from a set of options, reflecting more realistic, real-world scenarios where multiple answers may be correct.

SATA-BENCH encompasses a diverse set of tasks across six domains: reading comprehension, news classification, event detection, toxicity identification, biomedical concept tagging, and legal document analysis. The authors evaluate 30 state-of-the-art LLMs on this benchmark and reveal significant performance gaps, with even the best models achieving only 41.8% exact match accuracy. They identify two primary challenges contributing to this performance deficit: selection bias, where models favor certain choices regardless of content, and count bias, where models fail to predict the correct number of answers.

To address these issues, the authors propose Choice Funnel, an iterative decoding algorithm that combines token debiasing with adaptive thresholding. Choice Funnel systematically guides LLMs toward more accurate and complete selections, significantly improving exact match accuracy by up to 29% compared to competitive baselines, while also reducing inference costs.

**Dataset Code Accessibility:**

Yes

**Dataset Code Comments:**

The dataset is available.

**Ethical Considerations:**

No, there are no or only very minor ethics concerns

**Limitations Weaknesses:**

1. Limited Domain Coverage: While SATA-BENCH covers six domains, it still represents a narrow slice of possible applications. Additional domains such as mathematics, coding, or other specialized fields could enhance the benchmark's comprehensiveness and generalizability.

2. Potential for Memorization: As the dataset is derived from existing sources, there is a possibility that some LLMs, especially those trained on large-scale internet data, might have been exposed to parts of the dataset during training. This could affect the evaluation's validity in measuring true generalization.

3. Focus on Open-Source Models for Algorithm Evaluation: The effectiveness of Choice Funnel is demonstrated primarily on open-source models due to accessibility of token probabilities. It's unclear how well the algorithm would scale or perform with larger proprietary models where such probabilities might not be available.

4. Assumption of Token Debiasing Effectiveness: The Choice Funnel relies on token debiasing methods, but the adaptation of these methods from single-answer MCQ settings to SATA tasks may not fully address all underlying biases. Further investigation into alternative debiasing techniques or more robust bias mitigation strategies could strengthen the approach.

**Strengths Contributions:**

I love that this paper introduce a new Benchmark, the first of its kind benchmark focused on evaluating LLMs on SATA-type questions, considering many real-world applications require selecting multiple correct answers.

1. Diverse Dataset: SATA-BENCH includes 1,604 human-validated questions spanning various domains and difficulty levels.

2. Comprehensive Evaluation of LLMs: The authors conduct an extensive evaluation of 30 open-source and proprietary LLMs.

3. Proposal of the Choice Funnel Algorithm: By integrating token debiasing and adaptive thresholding, it enhances model performance on SATA tasks while reducing computational costs.

4. Open Availability: The authors provide access to the SATA-BENCH dataset and the Choice Funnel code, promoting transparency and enabling further research in this area.

---

> ### Author Rebuttal · Authors · 2025-07-30
>
> Thank you for your thorough and constructive review. We sincerely appreciate your recognition that SATA-BENCH is the *first benchmark specifically focused on evaluating LLMs on SATA-style (multi-answer) questions*. Your acknowledgment of the dataset’s diversity, its public availability, our decoding algorithm, and the comprehensive evaluation is especially encouraging. Below, we address your concerns in detail.
>
>
> **W1:  represents a narrow slice of possible applications. Additional domains such as mathematics, coding, or other specialized fields could enhance the benchmark's comprehensiveness and generalizability.**
>
> - We designed our benchmark to reflect realistic applications that LLMs are likely to encounter. The six current domains were chosen based on practical relevance:
>     - LLM-assisted annotation: Tasks like content moderation, compliance checks, or topic classification often require assigning multiple labels. This motivates our inclusion of datasets like Toxicity, News, and Events.
>     - Education and self-study: In domains like law or medicine, users may ask LLMs which principles, diagnoses, or statutes apply to a scenario. (Law and Biomedicine datasets)
>     - Document summarization or retrieval: Extracting all relevant facts from text is a multi-answer task. (Reading Comprehension dataset)
>
> - These domains together provide broad cognitive and linguistic coverage. Moreover, SATA-BENCH complements existing benchmarks (which focus on single-answer formats or math/coding problems) by specifically targeting the **multi-answer reasoning challenge** in real applications.
> - We excluded math and coding from this benchmark since they are already well-studied, and exhaustive multi-answer selection is less commonly required in real-world coding/math QA settings. We agree that math and coding questions are essential for LLM evaluations and plan to expand to math and coding in future versions to further enhance generality.
>
> **W2: Potential for Memorization: there is a possibility that some LLMs, especially those trained on large-scale internet data, might have been exposed to parts of the dataset during training. This could affect the evaluation's validity in measuring true generalization.**
> - While we cannot entirely eliminate the possibility of memorization, we applied the open-source contamination detection pipeline from [1]. Using the Bing Search API, we tested top 20 relevant queries per question to check for verbatim web overlap. We then cross-referenced hits with Common Crawl indexes. No questions were flagged as contaminated, indicating that our data is neither indexed in Common Crawl nor retrievable via public search. This reduces the likelihood that any model saw our questions during pre-training.
>
>
>
> **W3: Focus on Open-Source Models for Algorithm Evaluation: The effectiveness of Choice Funnel is demonstrated primarily on open-source models due to accessibility of token probabilities. It's unclear how well the algorithm would scale or perform with larger proprietary models where such probabilities might not be available.**
>
> - **Our algorithm is effective even without token probabilities.** In Appendix M.3, we conduct an ablation with a “NOTA-only” variant of Choice Funnel, which removes the requirement for probability-based thresholding and relies solely on iterative decoding with the auxiliary “None of the above” (NOTA) option. Across three models, this variant increased EM by 10.79%, increased JI by 20.51%, reduced SPD by 13.4, and reduced CtAbsDif by 0.86 on average. These results demonstrate that our algorithm can improve accuracy and reduce both selection and count bias, even when probability scores are unavailable. This makes the NOTA-only variant a practical solution for proprietary models where token-level probabilities cannot be accessed.
>
>
> - **Choice Funnel demonstrates strong effectiveness in larger model.** To further assess scalability, we evaluated Choice Funnel on the larger LLAMA3.1-70B model. As shown below, Choice Funnel continues to deliver substantial improvements over prompting-only approaches:
> | Model | EM↑ | Recall↑ | SPD↓ | CtAcc↑ |
> |-------|------------------|--------|-----|------------------------|
> | LLAMA3.1-70B + prompting | 17.94 | 60.64% | 1.81 | 22% |
> | LLAMA3.1-7B + ChoiceFunnel | 19.88 | 56.19% | 7.75 | 33% |
> | LLAMA3.1-70B + ChoiceFunnel| **24.43** | **68.66%**  | **0.37**  | **37%** |
>
> - These results show that Choice Funnel scales well to larger models, and consistently outperforms standard prompting in both accuracy and bias reduction.
>
>
> **W4: Assumption of Token Debiasing Effectiveness: The Choice Funnel relies on token debiasing methods, but the adaptation of these methods from single-answer MCQ settings to SATA tasks may not fully address all underlying biases. Further investigation into alternative debiasing techniques or more robust bias mitigation strategies could strengthen the approach.**
>
> - We agree that token debiasing methods cannot address all underlying bias such as count and selection bias. We address this limitation by combining iterative decoding with token debiasing. As described in Appendix M.2, we performed ablation studies to isolate the contributions of each component:
>
> | Method           | EM↑   | Precision↑ | Recall↑ | JI↑   | SPD↓   | CtDifAbs↓ | CtAcc↑ |
> |------------------|-------|------------|---------|-------|--------|-----------|--------|
> | Debiasing only   | 5.48  | 63.27      | 32.98   | 29.97 | 158.72 | 2.34      | 0.11   |
> | Iterative decoding only          | 19.94 | 74.44      | 56.39   | 49.91 | 12.46  | **1.59**      | **0.35**   |
> | choice funnel (iterative decoding + debiasing )  | **23.11** | **76.19**      | **60.44**   | **52.79** | **9.66**   | 1.60      | 0.34   |
>
> - The results clearly show that neither debiasing nor iterative decoding alone fully resolves bias; however, their combination yields the best overall Exact Match and consistently improves other metrics.
> - To our knowledge, our work is the first to use both techniques for SATA tasks. We will use these techniques as a starting point for developing more robust bias mitigation strategies in multi-answer settings.
>
> We appreciate your insightful comments, which have helped us clarify these key points. We hope our additional analyses address your concerns and provide further evidence of the robustness and scalability of our approach. We will revise the manuscript to make the above points explicit.
>
> [1] Li, Yucheng, Frank Guerin, and Chenghua Lin. "An open source data contamination report for large language models." arXiv preprint arXiv:2310.17589 (2023).

---

> ### Author Response · Authors · 2025-08-05
>
> We sincerely appreciate the time and effort you have already dedicated to reviewing our work. As the discussion phase is approaching its end,, we would be grateful if you could take a moment to consider our responses. Please let us know if there are any remaining concerns or points that would benefit from further clarification or additional experimentation. We hope that our responses have addressed the issues raised, and we would welcome any additional feedback you might have.

---

### Note · Authors · 2025-08-12

We thank the reviewers for their thoughtful evaluations.  We are encouraged by the reviewers' recognition of SATA-BENCH summarized below:
1. The **Novel Benchmark** represents the first dedicated benchmark for SATA-type questions curated through a multi-stage transformation pipeline. Our work is the first to quantify and analyze selection bias in SATA-style questions by introducing the novel Unselection Bias and Count Bias, providing new metrics (SPD, CountAcc) for community use.
2. The **broad domain coverage** of our work has been positively acknowledged by three reviewers. We release 3 datasets with > 10k human audited questions.
3. Our **comprehensive evaluation of 30 LLMs** (both proprietary and open-source) on SATA tasks.
4. The **innovative Choice Funnel Algorithm** has demonstrated significant improvements in both performance and computational efficiency, reducing inference forward passes by up to 64.6% while improving exact match rate by 29%.

We are encouraged by the positive responses from Reviewers YYAZ and TePK regarding our resolution of their concerns. Below, we provide concise clarifications on Reviewer Ymcx’s points:

**Application Scope**: While focused, our benchmark targets high-impact, real-world multi-answer tasks, including LLM-assisted annotation, education, and document retrieval. These domains were chosen for their practical frequency and cognitive diversity, and future versions will expand to include mathematics and coding. Two other reviewers are satisfied with this answer.

**Memorization Concerns**:  Our verification using both Bing Search API on top 20 queries and Common Crawl indexes has confirmed the originality of our questions, with zero examples flagged as contaminated, addressing these concerns comprehensively. Reviewer TePK is satisfied with this answer.

**Choice Funnel Effectiveness**: The data strongly supports our algorithm's effectiveness across three models even without relying on token probability (see Appendix M.3 for details). On large model LLAMA 3.1 70B, Choice Funnel achieved 36.2% increase in exact match and 13.23% in recall over prompting method (see Ymcx W3).

**Token Debiasing Effectiveness**:
We agree that token debiasing alone is not enough in SATA. The Choice Funnel algorithm is better than token debiasing across all metrics. (see Appendix M.2 and Ymcx W4).

We appreciate Ymcx's thoughtful questions. We will revise the manuscript to include all points mentioned by all reviewers.

---

### Decision · Program_Chairs · 2025-09-18

**Decision:**

Reject

**Comment:**

**Summary**
The paper introduces SATA-BENCH, a benchmark specifically designed to evaluate large language models on "Select All That Apply" (SATA) multi-answer tasks across six domains including law, biomedicine, and reading comprehension. The authors evaluate 30 models and uncover significant weaknesses, with best models achieving only 41.8% exact match, driven by selection bias and count bias. To address these issues, they propose the Choice Funnel decoding algorithm, which integrates token debiasing and adaptive thresholding, achieving up to 29% improvement over baselines.

**Strengths of the paper**

* Proposes a novel benchmark targeting SATA tasks, filling an important gap in existing LLM evaluations that typically focus on single-answer multiple-choice questions.
* Provides a diverse, human-validated dataset (1,604 questions) across six domains, with transparent release of both dataset and code.
* Offers a comprehensive evaluation of 30 proprietary and open-source LLMs, revealing fundamental limitations (selection and count biases).
* Introduces the Choice Funnel algorithm, which systematically reduces these biases and improves exact match accuracy while lowering inference cost.

**Weaknesses of the paper**

* Dataset scale is relatively small (1,604 questions) and covers only six domains, limiting comprehensiveness and generalizability.
* Potential overlap with pretraining corpora may compromise true generalization; memorization cannot be fully ruled out.
* Dataset construction has inconsistencies and redundancies in labels, which may artificially amplify bias measurements and reduce the reliability of results.
* Evaluation of Choice Funnel is limited to open-source models (due to probability access), with insufficient analysis of why it sometimes fails or amplifies bias, and little discussion of factors like option length or random distractor sampling.

**Major reason to accept**
The paper proposes a novel benchmark targeting SATA tasks with a significant contribution to the field.

**Summary of rebuttal and discussions**
The reviewers initially raised concerns, but the authors have addressed all of them and hence reached a consensus of acceptance.

===== FINAL UPDATE FROM DB Track PCs ====

The final decision for this paper has been taken by the program chairs after consultation with the SACs. All Senior Area Chairs have ranked papers according to the feedback from the AC during the review process. We decided to leave the original meta-review to reflect the opinion of the AC in light of the initial discussions with reviewers and SAC.